# The ALT pathway generates telomere fusions that can be detected in the blood of cancer patients

Francesc Muyas [1,6], Manuel José Gómez Rodriguez[2,6], Rita Cascão[3], Angela Afonso[3], Carolin M. Sauer [1], Claudia C. Faria[3,4], Isidro Cortés-Ciriano [1] ✉ & Ignacio Flores [2,5] ✉

Telomere fusions (TFs) can trigger the accumulation of oncogenic alterations leading to malignant transformation and drug resistance. Despite their relevance in tumour evolution, our understanding of the patterns and consequences of TFs in human cancers remains limited. Here, we characterize the rates and spectrum of somatic TFs across >30 cancer types using whole-genome sequencing data. TFs are pervasive in human tumours with rates varying markedly across and within cancer types. In addition to end-to-end fusions, we find patterns of TFs that we mechanistically link to the activity of the alternative lengthening of telomeres (ALT) pathway. We show that TFs can be detected in the blood of cancer patients, which enables cancer detection with high specificity and sensitivity even for early-stage tumours and cancers of high unmet clinical need. Overall, we report a genomic footprint that enables characterization of the telomere maintenance mechanism of tumours and liquid biopsy analysis.

Telomeres are nucleoprotein complexes composed of telomeric TTAGGG repeats and telomere binding proteins that prevent the recognition of chromosome ends as sites of DNA damage[1]. The replicative potential of somatic cells is limited by the length of telomeres, which shorten at every cell division due to end-replication losses. Most human cancers maintain the length of telomeres above a critical threshold by re-expressing telomerase through diverse mechanisms[2], including activating *TERT* promoter mutations[3,4] and enhancer hijacking[5]. In other cancers, in particular those of mesenchymal and neuroendocrine origin, telomeres are elongated by the alternative lengthening of telomeres (ALT) pathway, which relies on recombination[6]. Telomere attrition can result in senescence or the ligation of chromosome ends to form dicentric chromosomes, which are observed as chromatin bridges during anaphase[7]. The resolution of chromosome bridges caused by telomere fusions (TFs) can increase

genomic instability and the acquisition of oncogenic alterations involved in malignant transformation and resistance to chemotherapy through diverse mechanisms, including chromothripsis and breakage-fusion-bridge cycles[8–12].

Despite their importance to tumour evolution, the patterns and consequences of TFs remain largely uncharacterized, in part due to technical challenges. TFs have been traditionally detected by inspection of chromosome bridges in metaphase spreads, an approach that is not scalable and has low resolution[13–15]. In recent years, the study of TFs has relied on PCR-based methods using primers annealing to subtelomeric regions[16,17], which are limited to detect TFs distantly located from subtelomeres since PCR efficiency decreases as the amplicon size increases[18]. To overcome these limitations, we have developed computational methods to detect TFs using whole-genome sequencing (WGS) data[19]. Here, we report the rates and patterns of TFs across

[1]European Molecular Biology Laboratory, European Bioinformatics Institute, Hinxton, Cambridge CB10 1SD, UK. [2]Centro Nacional de Investigaciones Cardiovasculares Carlos III (CNIC), Madrid 28029, Spain. [3]Instituto de Medicina Molecular João Lobo Antunes, Faculdade de Medicina, Universidade de Lisboa, Lisboa, Portugal. [4]Department of Neurosurgery, Hospital de Santa Maria, Centro Hospitalar Universitário Lisboa Norte (CHULN), Lisboa, Portugal. [5]Centro de Biologia Molecular Severo Ochoa, CSIC-UAM, Cantoblanco, Madrid 28049, Spain. [6]These authors contributed equally: Francesc Muyas, Manuel José Gómez Rodriguez. ✉e-mail: icortes@ebi.ac.uk; iflores@cnic.es

>2,000 tumours spanning >30 cancer types. In addition, we report the discovery of a type of TF which we mechanistically link with the ALT pathway. We also demonstrate that ALT and non-ALT tumours generate TFs that can be detected in the blood of cancer patients with high sensitivity and specificity for liquid biopsy analysis.

## Results

### Pan-cancer landscape of telomere fusions

To detect TFs in sequencing data, we developed TelFusDetector[20] (Fig. 1A). In brief, TelFusDetector identifies sequencing reads containing at least two consecutive TTAGGG and two consecutive CCCTAA telomere sequences, allowing for mismatches to account for the variation observed in telomeric repeats in humans[21,22] (Fig. 1, Supplementary Fig. 1; see Methods for a detailed description of the algorithm, including QC steps). We first scanned the human reference genome to identify regions containing telomere fusion-like patterns, which could be misinterpreted as somatic. Our analysis revealed the relic of an ancestral fusion in chromosome 2[23], and a region in chromosome 9 containing 2 sets of telomeric repeats flanked by high complexity sequences, which we term "chromosome 9 endogenous fusion" (Supplementary Table 1). To assess the performance of TelFusDetector, we simulated telomeric fusions using the T2T-CHM13 assembly across different read depths (Methods and Supplementary Fig. 2). The sensitivity of TelFusDetector ranged from 0.66 to 0.90 for simulated read depths of 1x and 40x, respectively. Notably, TelFusDetector did not detect any false telomere fusions, indicating the high precision of our approach (Supplementary Fig. 2).

To characterize the patterns and rates of somatic TFs across diverse cancer types, we applied TelFusDetector to 2071 matched tumour and normal sample pairs from the Pan-Cancer Analysis of Whole Genomes (PCAWG) project that passed our QC criteria (Methods, Supplementary Fig. 1). To enable comparison of the relative number of TFs across samples, we computed a TF rate for each tumour after correcting for tumour purity, sequencing depth, and read length (Fig. 1B, Supplementary Fig. 3, Supplementary Data 1). In silico simulations showed that correcting for tumour purity is required to compare telomere fusion rates across tumours of variable tumour cellularity (Methods and Supplementary Fig. 2). We identified two distinct TF patterns, which differ in the relative position of the sets of TTAGGG and CCCTAA repeats (Fig. 1A). A first pattern, which we term "inward TF", is characterized by 5'-TTAGGG-3' repeats followed by 5'-CCCTAA-3' repeats, which is the expected genomic footprint of end-to-end TFs[1,13]. Unexpectedly, we also found a second pattern characterized by 5'-CCCTAA-3' repeats followed by 5'-TTAGGG-3' repeats, which we term "outward TF" (Fig. 1A). In addition, we found read pairs where a read in the pair contained an inward TF and the other an outward TF, which would be consistent with a circular DNA element, and thus we term these circular (in-out) TFs.

Both outward and inward TFs were detected across diverse cancer types, but rates varied markedly within and across tumour types (Fig. 1B). The highest TF rates were observed in osteosarcomas (Bone-Osteosarc), leiomyosarcomas (SoftTissue−Leiomyo), and pancreatic neuroendocrine tumours (Panc−Endocrine). The lowest frequencies were observed in thyroid adenocarcinomas (Thy−AdenoCA), renal cell-carcinomas (Kidney−RCC), and uterine adenocarcinomas (Uterus−AdenoCA). These results indicate that somatic TFs, including the outward telomere fusions we report here, are pervasive across diverse cancer types.

### The ALT pathway is mechanistically linked with the formation of telomere fusions

Next, we sought to determine the molecular mechanisms implicated in the generation of TFs. To this aim, we regressed the observed rates of TFs on the mutation status of *ATRX* and *DAXX*, telomere content, point

mutations and structural variants in the *TERT* promoter, expression values of *TERT* and TERRA, and a binary category indicating the ALT status of each tumour predicted using two previously published classifiers[21,24]. Our analysis revealed a strong association between the activation of the ALT pathway and the rate of TFs, with the strongest effect size observed for outward TFs and *TP53* inactivation ($P < 0.05$; Fig. 2A-B, Supplementary Fig. 4A). However, alterations of the *TERT* promoter were negatively correlated with both inward and outward TF rates, with the highest effect size obtained for outward TFs (Fig. 2A). The association of TF rates with telomere content and TERRA expression was also significant, although of a modest effect size ($P < 0.001$, Supplementary Fig. 5).

To investigate the association between the ALT pathway and TF formation, we first compared the rate of TFs between tumours positive and negative for C-circles, an ALT marker[21,24]. For this analysis, we focused on published data from 42 skin melanomas and 53 pancreatic neuroendocrine tumours that are also part of the PCAWG cohort. ALT tumours showed significantly higher rates of TFs in the pancreatic neuroendocrine tumour set ($P < 0.001$, two-tailed Mann-Whitney test; Fig. 2C). A similar trend was observed for skin melanomas, although only outward fusion rates reached significance (Fig. 2C). Next, we extended this analysis to the entire cohort by comparing the TF rates between tumours with high and low ALT-probability scores (ALT-low vs ALT-high)[24] on a per cancer type basis (Fig. 2D). Overall, TF rates, in particular outward TFs, were significantly higher in cancer types classified as ALT-high (FDR-corrected $P < 0.1$, two-tailed Mann-Whitney test; Fig. 2D, Fig. 1B).

To test whether TF fusions are enriched in ALT cancers, we analysed whole-genome sequencing data from 306 cancer cell lines from the Cancer Cell Line Encyclopedia (CCLE)[25]. Consistent with our observations in primary tumours, cell lines used as models of ALT, such as the osteosarcoma cell line U2OS and the melanoma cell line LOXIMVI, showed the highest rates of both inward and outward TFs (Fig. 2E, Supplementary Fig. 1I). Analysis of PacBio long-read sequencing data from the ALT breast cancer cell line SK-BR-3[26] also revealed an enrichment of outward TFs in this line as compared to the non-ALT cell lines COLO289T, HCT116, KM12, SW620 and SW837, for which long-read sequencing data were also available[27,28] (Supplementary Fig. 4B, C, Supplementary Fig. 6).

To assess whether TFs are specifically associated with the ALT pathway, we analysed the genomes of mortal cell strains before and after transformation by mechanisms requiring telomerase or ALT[29]. The genomes of parental mortal strains JFCF-6 and GM02063, as well as telomerase-positive strains JFCF-6/T.1F and GM639, did not contain outward TFs (Fig. 2F). In contrast, ALT-derived strains JFCF-6/T.1R, JFCF-6/T.1M and GM847 showed a comparable outward TF fusion rate to the prototypical ALT cell line U2OS (Fig. 2F). Therefore, the presence of outward TFs in ALT-derived-strains but not in telomerase-positive strains indicates that ALT activation leads to the formation of outward TFs. In addition, we analysed TF rates in whole-genome sequencing data from hTERT-expressing retinal pigment epithelial (RPE-1) cells sequenced after the induction of telomere crisis using a dox-inducible dominant negative allele of *TRF2*[9,10]. Compared to the control samples sequenced before induction of telomere crisis, we detected a high rate of inward TFs, which is consistent with the presence of end-to-end fusions ($P < 0.05$, two-tailed Mann-Whitney test; Fig. 2G, Supplementary Fig. 1J), thus lending further support to the mechanistic association between ALT activity and the formation of outward TFs. Consistent with the activation of the ALT pathway in cells immortalized by the Epstein-Barr virus in vitro[30,31], we also detected high rates of outward TFs in 2490 Epstein−Barr virus-immortalized B cell lines from the 1000G project (Fig. 2H, Supplementary Fig. 1).

To further test the association between TFs and ALT activity, we used Random Forest classification to predict the ALT status of tumours

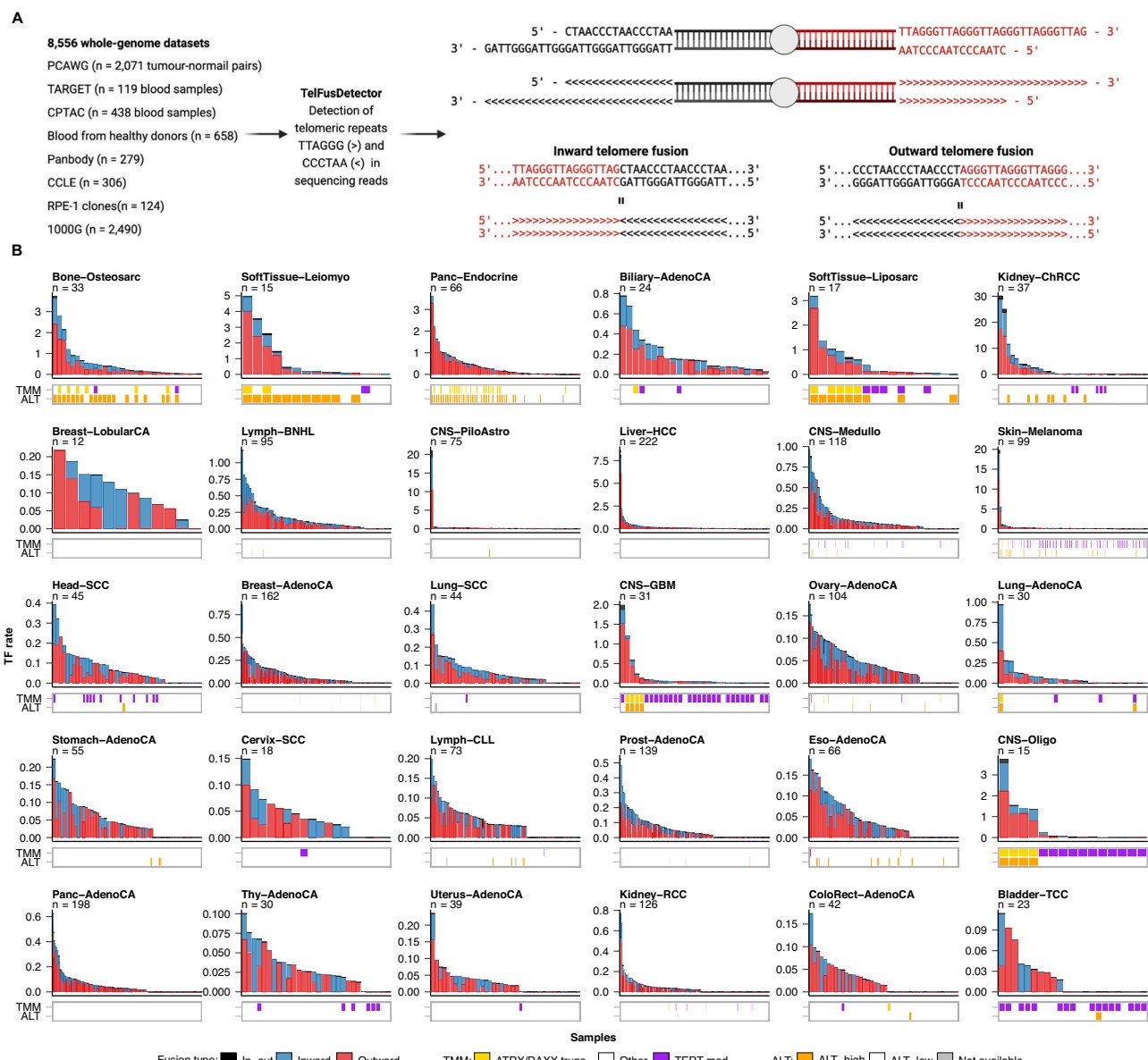

**Fig. 1 | Landscape of telomere fusions in cancer. A** Overview of the study design and schematic representation of the two types of telomere fusions (TFs) identified by TelFusDetector. **B** TF rates per sample across 30 cancer types from PCAWG. Each bar corresponds to one patient and the bars are ordered decreasingly by the TF rates. Outward fusions are shown in red and inward in blue. Read pairs where a read in the pair contained an inward TF and the other an outward TF, which would be consistent with a circular DNA element, are indicated as in-out TFs and shown in grey. The colour of the boxes below the bar plots represents the telomere maintenance mechanism (TMM) predictions reported by de Nonneville and Reddel, 2021 and the TMM mutational status reported by Sieverling et al. 2020. Only cancer types with at least 10 tumours are shown. ATRX/DAXXtrunc, tumours with inactivating point mutations, frameshift indels and structural variants in *ATRX* or *DAXX*; TERT mod., tumours with activating mutations in *TERT*. The abbreviations used for the cancer types are as follows: Biliary-AdenoCA biliary adenocarcinoma, Bladder-TCC bladder transitional cell carcinoma, Bone-Benign bone cartilaginous neoplasm, osteoblastoma and bone osteofibrous dysplasia, Bone-Epith bone neoplasm, epithelioid, Bone-Osteosarc sarcoma, bone, Breast-AdenoCA breast adenocarcinoma, Breast-DCIS breast ductal carcinoma in situ, Breast-LobularCA breast lobular carcinoma, Cervix-AdenoCA cervix adenocarcinoma, Cervix-SCC cervix squamous cell carcinoma, CNS-GBM central nervous system glioblastoma, CNS-Oligo CNS oligodenroglioma, CNS-Medullo CNS medulloblastoma, CNS-PiloAstro CNS pilocytic astrocytoma, ColoRect-AdenoCA colorectal adenocarcinoma, Eso-AdenoCA esophagus adenocarcinoma, Head-SCC head-and-neck squamous cell carcinoma, Kidney-ChRCC kidney chromophobe renal cell carcinoma, Kidney-RCC kidney renal cell carcinoma, Liver-HCC liver hepatocellular carcinoma, Lung-AdenoCA lung adenocarcinoma, Lung-SCC lung squamous cell carcinoma, Lymph-CLL lymphoid chronic lymphocytic leukemia, Lymph-BNHL lymphoid mature B-cell lymphoma, Lymph-NOS lymphoid not otherwise specified, Myeloid-AML myeloid acute myeloid leukemia, Myeloid-MDS myeloid myelodysplastic syndrome, Myeloid-MPN myeloid myeloproliferative neoplasm, Ovary-AdenoCA ovary adenocarcinoma, Panc-AdenoCA pancreatic adenocarcinoma, Panc-Endocrine pancreatic neuroendocrine tumour, Prost-AdenoCA prostate adenocarcinoma, Skin-Melanoma skin melanoma, SoftTissue-Leiomyo leiomyosarcoma, soft tissue, SoftTissue-Liposarc liposarcoma, soft tissue, Stomach-AdenoCA stomach adenocarcinoma, Thy-AdenoCA thyroid low-grade adenocarcinoma, and Uterus-AdenoCA uterus adenocarcinoma. The ALT-TF rate indicates the number of ALT-TFs per 1x genome coverage.

using the rates and features of TFs as covariates, and the set of tumours with C-circle assay data as the training set. Variable importance analysis using the best performing classifier (AUC = 0.93) identified variables encoding the rate and breakpoint sequences of TFs as the most predictive features, followed by the fraction of telomere variant repeats (TVR) GTAGGG and CCCTAG, which are enriched in ALT tumours[32], in sequencing reads with TFs (Supplementary Fig. 7 and Methods).

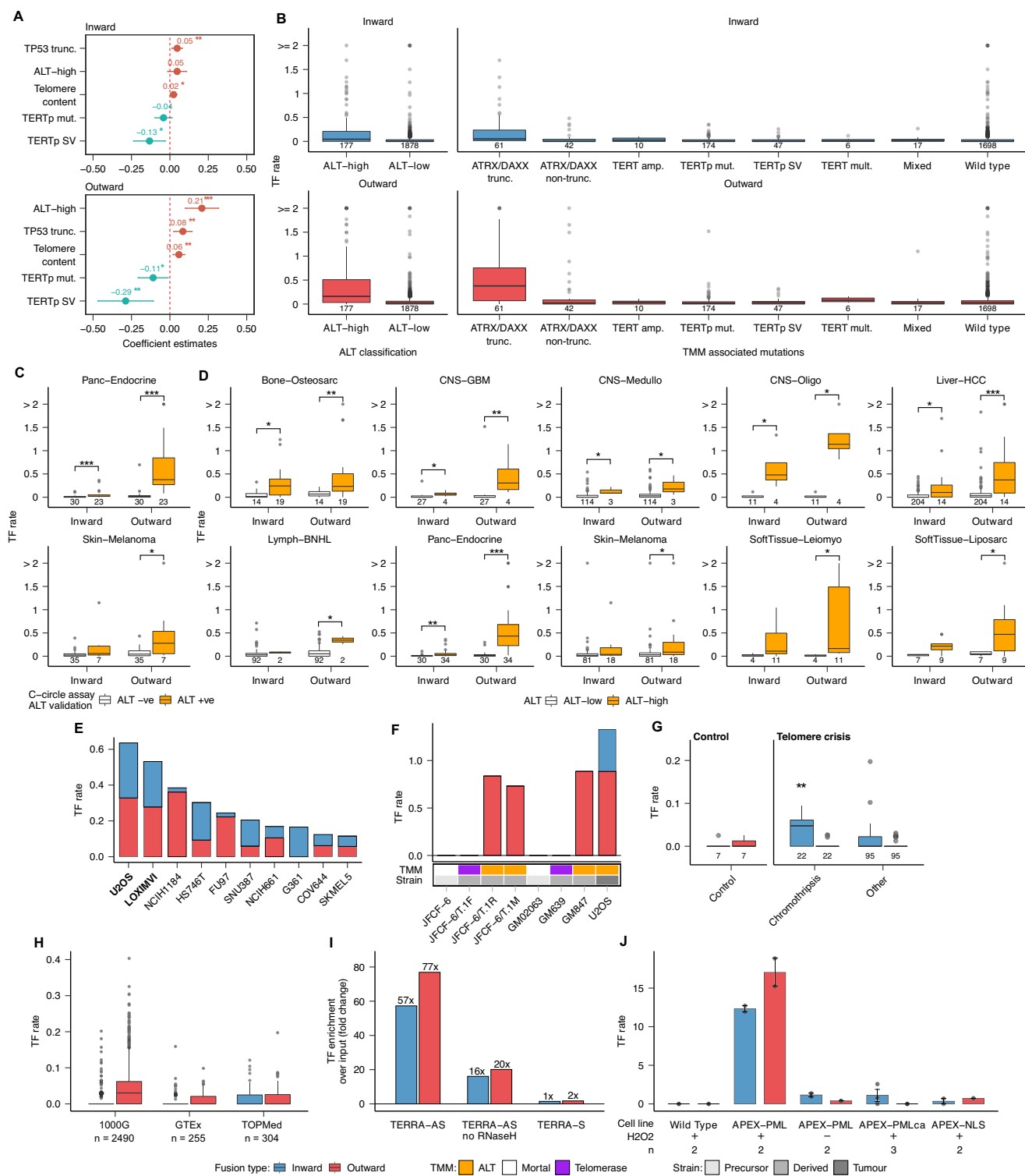

Together, these results mechanistically link the activity of the ALT pathway with the generation of somatic TFs. Therefore, we term inward and outward fusions ALT-associated TFs (ALT-TFs).

## ALT-TFs are found in TERRA-binding sites and APBs

We next sought to determine the association of ALT-TFs with molecules involved in telomere maintenance and their cellular localization. Our regression expression analysis of the PCAWG data set indicates that tumours enriched in ALT-TFs present elevated levels of TERRA, a long non-coding RNA transcribed from telomeres[33,34]. Previous genomic and cytological studies demonstrated a preferential association of

TERRA transcripts to telomeres[35]. To assess whether TERRA also associates with ALT-TFs, we searched for inward and outward ALT-TFs in reads containing TERRA-binding sites. Specifically, we analysed reads from CHIRT-seq, an immunoprecipitation protocol that specifically captures TERRA-binding sites using an anti-sense biotinylated TERRA transcript (TERRA-AS) as bait[36]. Targets of the TERRA-AS bait are then treated with RNase H to elute DNA containing TERRA binding sites followed by sequencing. By analysing CHIRT-seq data sets from mouse embryonic stem cells (mESCs)[36], a cell type with prominent TERRA foci[37], we observed a 57-fold and 77-fold enrichment of inward and outward ALT-TFs, respectively, over the input using the TERRA-AS

**Fig. 2 | TFs are generated by the activity of the ALT pathway. A** Coefficient values estimated using linear regression analysis and variable selection for the covariates with the strongest positive and negative association with TF rates. For this analysis we used the ALT status classification reported by de Nonneville and Reddel, 2021. Only samples with complete information for all variables used in the linear regression analysis were included (n = 2050). Dots represent the estimated coefficient values and bars show the 95 % confidence interval. **B** Rates of inward and outward TFs in PCAWG tumours grouped by ALT status predictions (de Nonneville and Reddel, 2021), and TMM-associated mutations (Sieverling et al., 2020). **C** Comparison of TF rates between tumours positive and negative for the C-circle assay. *C-circles*: partially single-stranded telomeric (CCCTAA)n DNA circles, known to be an ALT specific biomarker. **D** TF rates estimated for PCAWG tumours grouped by ALT status predictions across selected cancer types. Only cancer types with significant differences in the number of TFs based on ALT status are shown. **E** Top 10 cancer cell lines from the CCLE with the highest TF rates. ALT cell lines are indicated in bold type. **F** TF rates in mortal cell strains before and after transformation by mechanisms requiring telomerase or ALT. **G** TF rates detected in RPE-1 cell lines before (control) and after induction of telomere crisis with doxycycline. **H** TF rates detected in 1000G, GTEx and TOPMed samples. **I** Fold changes in TF rates over the control estimated using CHIRT-seq data from mouse embryonic stem cells treated with TERRA-AS, TERRA-AS without Rnase H, and TERRA-S. **J** TF rates estimated using AlaP data generated using different conditions of APEX knock-in and peroxidase (H2O2). The bars represent the mean TF rate, and the error bars show the 95 % confidence interval. *APEX-PMLca*: ALaP-seq for the PML RING domain mutant (C62A/C65A [PMLca]), which diffusively localizes in the nucleoplasm without forming PML bodies; *APEX-NLS*: An APEX negative control used as a freely distributed nuclear protein. In all panels ***P < 0.001; **P < 0.01; *P < 0.05, Wilcoxon rank-sum test after FDR correction. Box plots show the median, first and third quartiles (boxes), and the whiskers encompass observations within 1.5x the interquartile range from the first and third quartiles. ATRX/DAXX trunc., tumours with inactivating point mutations, frameshift indels and structural variants in *ATRX* or *DAXX*; ATRX/DAXX non-trunc., tumours without inactivating mutations in *ATRX* or *DAXX*; TERT amp., tumours with at least 6 additional *TERT* copies over the tumour ploidy; TERTp mut., tumours with activating mutations in the *TERT* promoter region; TERTp SV, a structural variant breakpoint on the plus strand 20 kb upstream of *TERT*; TERT mult., tumours with multiple *TERT* alterations. Mixed, tumours with both an activating *TERT* alteration and an *ATRX/DAXX* alteration. Numbers under the boxplots in (**B**–**D**–**G**) represent the number of samples included in each group.

oligo probe (Fig. 2I). However, a modest enrichment was observed when the TERRA-AS was not treated with RNase H or the TERRA sense transcript (TERRA-S) was used (Fig. 2I). These results thus indicate that TERRA binds to inward and outward ALT-TFs.

TERRA transcripts can be found in a subtype of promyelocytic leukaemia nuclear bodies (PML-NB) termed ALT-associated PML-Bodies (APBs)[38]. Because ALT-TFs bind to TERRA, we hypothesized that inward and/or outward ALT-TFs might locate to APBs. Given that PML-NBs, including APBs, are insoluble[39], a standard ChIP-seq protocol for PML cannot be used to analyse whether ALT-TFs are present in APBs. To overcome PML-NB accessibility problems, Kurihara et al.[40] recently developed an assay called ALaP, for APEX-mediated chromatin labelling and purification by knocking in APEX, an engineered peroxidase, into the *Pml* locus to tag PML-NB partners in an H2O2-dependent manner. Applying ALaP in mESCs, PML-NBs bodies were found to be highly enriched in ALT-related proteins, such as DAXX and ATRX, as well as in telomere sequences[40]. Here, to test our hypothesis, we searched for ALT-TFs in ALaP genomic pull-downs and found a strong enrichment of both inward and outward ALT-TFs (P < 0.05, two-tailed Mann-Whitney test; Fig. 2J). In addition, we found that negative controls, *i.e.*, APEX-PMLs not-treated with H2O2 or APEX variants that do not form PML-NBs, rarely contain ALT-TFs (Fig. 2J). Therefore, these results indicate that APBs are a preferential location for ALT-TFs. Furthermore, these findings indicate the presence of ALT-TF within mESCs, a cell type that not only exhibits telomere activity but possesses the capacity to elongate its telomeres through a telomere sister chromatid exchange (T-SCE) mechanism[41].

### Short DNA fragments contain ALT-TFs
In addition to APBs, another feature of ALT positive cells is their elevated levels of extrachromosomal telomeric DNA (ECT-DNA). Interestingly, most ECT-DNAs in ALT-positive cells localize to APBs[42]. As ALT-TFs also localize to APBs, it is conceivable that ECT-DNAs exert as substrates for the formation of ALT-TFs. If this was the case, the formation of ALT-TFs would result from the fusion of short ECT-DNA fragments rather than chromosomes. To test this hypothesis, we inferred the fragment size for read pairs with ALT-TFs or chromosome 9 endogenous fusions in which both mates support the same breakpoint sequence. We found a significant enrichment of ALT-TFs in DNA fragments shorter than the average library insert size in a set of cancer types with high ALT-TF rates, such as melanomas, osteosarcomas, and glioblastomas (FDR-corrected P < 0.1; Chi-square test; Supplementary Data 2). Complementary analyses performed on mESC PML-NB

sequencing data[40] indicated that 26% of ALT-TF-containing reads were significantly shorter than the average fragment length, while this happened for around 4% of the reads mapping outside telomeres. In addition, ALT-TF-containing fragments were three times shorter than the average fragment size in the library (Supplementary Fig. 8). Together, these results indicate that ALT-TFs might originate from the fusion of small DNA fragments.

### Sequence specificity at the telomere fusion point
We next analysed the set of sequences at the fusion point in ALT-TFs detected in PCAWG tumours. ALT-TFs with breakpoint sequences in the set of all possible circular permutations of TTAGGG and CCCTAA sequences were classified as pure (59% of ALT-TFs), whereas ALT-TFs with complex breakpoint sequences not comprising telomere reads were classified as alternative (41%; Supplementary Fig. 9, Supplementary Data 3). In pure ALT-TFs, we detected the entire set of possible permutations of telomere repeat motifs at fusion breakpoints, but not at similar frequencies (P < 0.05; chi-square test; Fig. 3A). In the case of outward ALT-TFs, the breakpoint sequence[5']...CCCTAA**CCCTAGGG**TTAGGG...[3'] was the most abundant (22% of pure ALT-TFs) followed by[5']...CCCTAA**CCC**TTAGGG...[3'] (16%). Interestingly, these two breakpoint sequences can be generated by the ligation of 7 and 4 combinations of double-strand telomeric repeats, respectively, while the other breakpoint sequences detected can only be generated by the combination of two specific telomeric repeat sequences (Fig. 3A, Supplementary Fig. 10). In addition, these two sequences are the only ones in the entire set of breakpoint sequences in outward ALT-TFs with microhomology at the fusion point. In the case of inward ALT-TFs, the[5']...TTAGGG**TTAA**CCCTAA...[3'] sequence was the most abundant (14% of pure ALT-TFs) followed by[5']...TTAGGG**TAA**CCCTAA...[3'] (14%). We also detected the TTAGCTAA sequence in 7% of pure ALT-TFs, which could be generated by the end-to-end fusion of two telomeres considering that ...TTAG[3'] is the most common terminal sequence at human telomeres[1]. Similar to outward ALT-TFs, TTAA and TAA are the only breakpoint sequences in inward fusions that can be created by the ligation of several combinations of telomeric repeats and contain microhomology at the fusion junction (Fig. 3A). We found a significant enrichment of ALT-TFs with microhomology at the breakpoint sequencing in inward and outward ALT-TFs (P < 0.001; Fig. 3B). As microhomology facilitates ligation, we conclude that microhomology at the fusion point also contributes to explain differences in the frequency of specific breakpoint sequences in both outward and inward ALT-TFs.

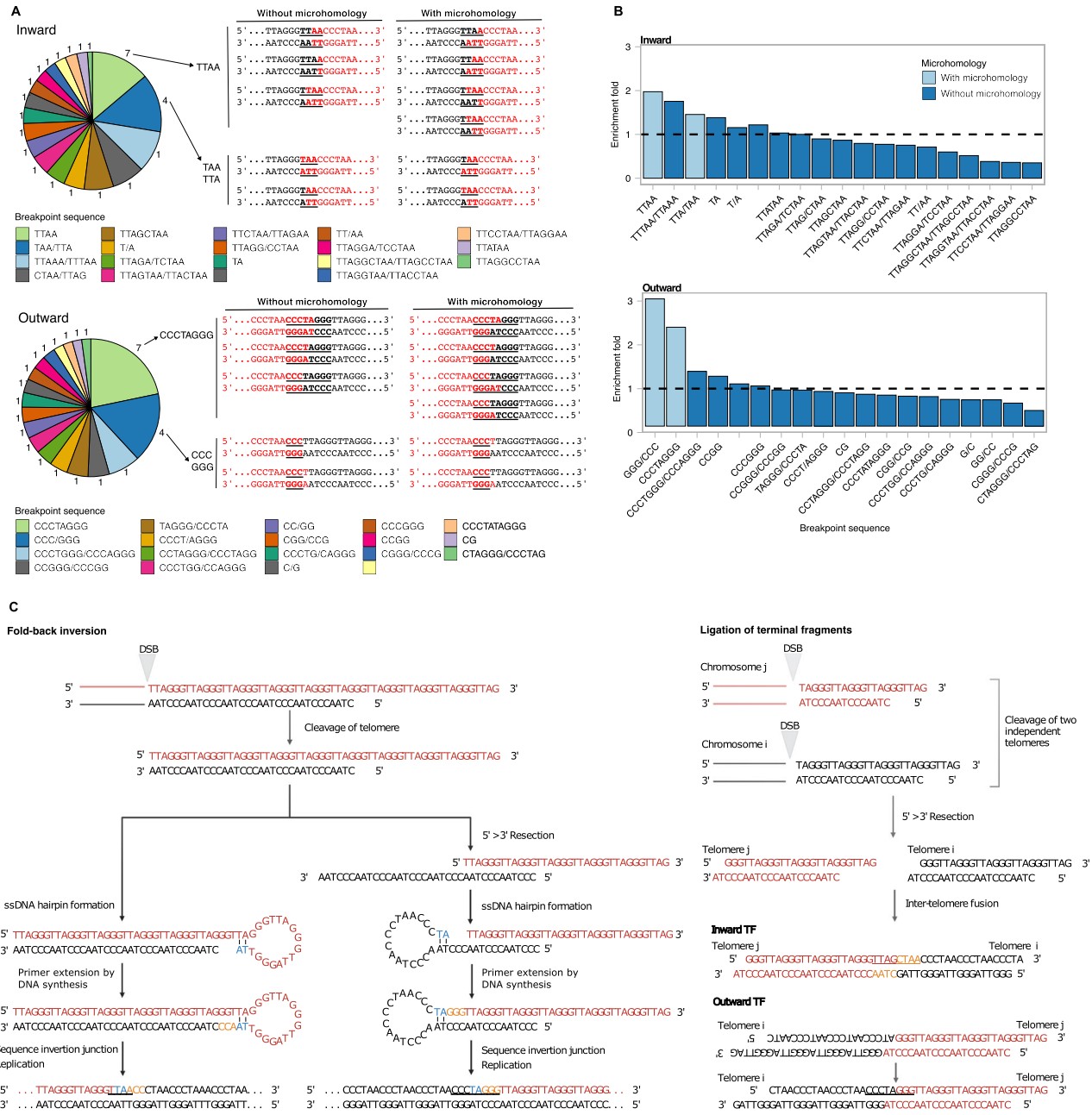

**Fig. 3 | Mechanisms of ALT-TF formation. A** Pie chart showing the distribution of the distinct breakpoint sequences observed in pure ALT-TFs in PCAWG tumours. The numbers around the pie charts represent the number of combinations of circular permutations of double-strand telomeric repeat motifs that can generate each breakpoint sequence. The legend reports the breakpoint sequences in both strands unless they are identical (e.g., TTAA). The most represented breakpoint sequences are indicated. Black and red letters, bold and plain fonts are used to highlight breakpoint sequences. **B** Enrichment of breakpoint sequences for inward and outward pure ALT-TFs. The colour indicates the presence or absence of microhomology at the fusion junction. **C** Proposed mechanisms for ALT-TF formation. ALT-TFs are generated through an intra-telomeric fold-back inversion after a double-strand break (DSB; left), or by the ligation of terminal telomere fragments after double-strand breaks (right).

## ALT-TFs are generated through the repair of double-strand breaks by an intra- or an inter-telomeric mechanism

Our previous analysis suggests that ALT-TFs are generated at APBs preferentially when telomeric fragments with microhomology in their ends fuse. Therefore, we postulate two non-exclusive mechanisms of ALT-TF formation (Fig. 3C, Supplementary Fig. 11). First, a double-strand break in a telomere can be repaired through an intra-telomeric fold-back inversion[43–45]. Specifically, end resection of a double-strand break would facilitate the formation of a hairpin loop when the 3' end of a telomere strand folds back to anneal its complementary strand through microhomology. Then, DNA synthesis would fill the gap to complete the capping of the hairpin. Finally, replication of the hairpin would create an inward or an outward ALT-TF depending on the 3' end telomeric strand that folds back: ...(TTAGGG)$_n$...$^{3'}$ fold-back would create an inward ALT-TF and ...(CCCTAA)$_n$...$^{3'}$ fold-back would create an outward ALT-TF. TRF2, a shelterin protein may mediate replication of the telomeric hairpin by recruiting the origin recognition complex (ORC)[46–48]. Secondly, ALT-TFs can also be generated through the ligation of the terminal fragments upon double-strand DNA breaks in telomeres (Fig. 3C, Supplementary Fig. 11). Specifically, an inter-

telomeric mechanism would occur when two telomeres covalently fuse in $^{5'}...(TTAGGG)_n...$-$...(CCCTAA)_n...^{3'}$ orientation to create an inward ALT-TF, or in $^{5'}...(CCCTAA)_n...$-$...(TTAGGG)_n...^{3'}$ orientation to create an outward ALT-TF. Outward ALT-TFs are only feasible when telomeric fragments join from the broken ends produced after telomere trimming (Fig. 3C).

## ALT-TFs are detected in blood and enable cancer detection

Given the high rate of ALT-TFs observed in tumours of diverse origin, we hypothesized that ALT-TFs could also be detected in blood samples and used as biomarkers for liquid biopsy analysis. To test this hypothesis, we first applied TelFusDetector to whole blood samples from PCAWG ($n = 1604$), the Genotype-Tissue Expression (GTEx; $n = 255$) project and the Trans-Omics for Precision Medicine program (TOPMed; $n = 304$). Overall, blood samples from cancer patients showed a significantly higher rate of ALT-TFs, especially of the outward type (FDR-corrected $P < 0.1$, two-tailed Mann-Whitney test; Fig. 4A and Supplementary Fig. 12).

Next, we utilized Random Forest (RF) classification to model the probability that an individual has cancer based on the patterns of ALT-TFs detected in blood (Methods). For this analysis, we also included blood samples from 438 cancer patients from the Clinical Proteomic Tumor Analysis Consortium (CPTAC) cohort, 119 pediatric cancer patients from The Therapeutically Applicable Research to Generate Effective Treatments (TARGET) program, and 99 healthy individuals from Korean Personal Genome Project (KPGP)[49]. In

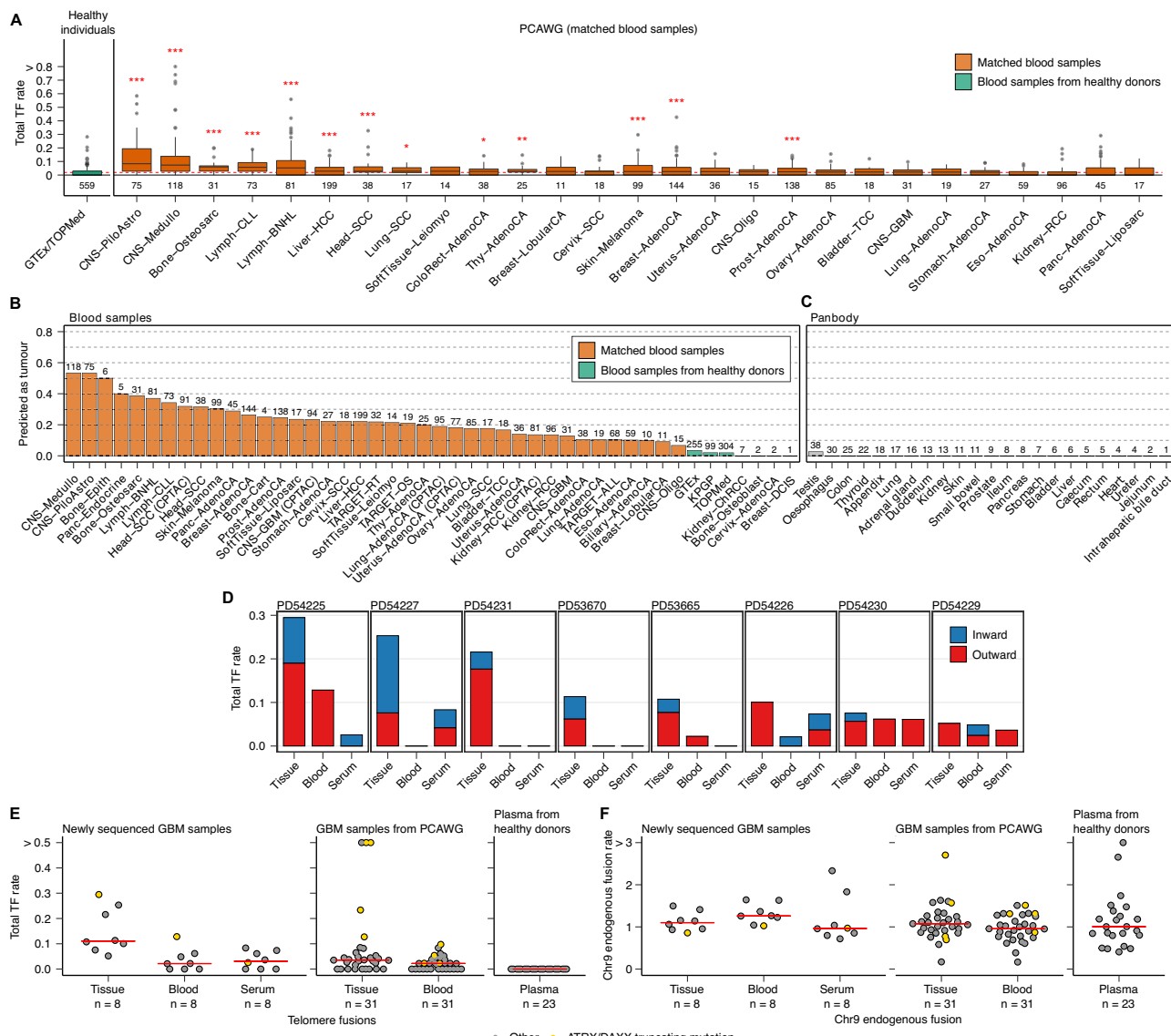

**Fig. 4 | ALT-TFs detected in blood and cell-free DNA enable cancer detection.**
**A** ALT-TF rates in blood samples from healthy individuals from GTEx and TOPMed (green) and blood samples from the PCAWG cohort (orange). **B** Fraction of individuals predicted to have cancer based on the patterns of ALT-TFs detected in blood. Predictions were computed using 100 Random Forest models trained on features of the ALT-TFs detected in WGS data from matched blood samples from CPTAC, PCAWG and TARGET. WGS data from blood samples from GTEx, KPGP and TOPMed were used as controls for model training. Only samples with at least 1 ALT-TF in blood were used for training. **C** Fraction of histologically normal samples from the Panbody study predicted as cancer using the same model. **D** ALT-TF rates

detected in the tumour tissue, whole blood and serum cell-free DNA from 8 GBM patients. **E** Comparison of ALT-TF rates detected in the GBM samples from different datasets, as well as in plasma cell-free DNA from healthy donors (upper plot).
**F** Comparison of chromosome 9 fusion rates detected in the GBM samples from different datasets, as well as in plasma from healthy donors. The numbers below the boxplots in (**A**) and above the bars in (**B**, **C**–**E**, **F**) indicate the total number of samples included in each group. Box plots in (**A**) show the median, first and third quartiles (boxes), and the whiskers encompass observations within a distance of 1.5x the interquartile range from the first and third quartiles. ***$P < 0.001$; **$P < 0.01$; *$P < 0.05$, Wilcoxon rank-sum test after FDR correction.

addition, we analysed 279 non-neoplastic samples spanning 29 histological units from the Panbody study[50]. In brief, each blood sample, from either a healthy donor or a cancer patient, was encoded by a vector recording 117 features of the ALT-TFs detected (Supplementary Data 4). The final RF predictions revealed high sensitivity for medulloblastomas (53%; Supplementary Fig. 13), pilocytic astrocytomas (53%), bone neoplasm epithelioid (50%) and pancreatic adenocarcinomas (40%) (Fig. 4B). These values were notably higher when only considering blood samples with at least 1 ALT-TF and remained comparable across cancer stages (Supplementary Fig. 12B–D). In contrast, the false positive rate was low in all control samples (2-4%) and even lower for samples from the Panbody study (Fig. 4C)[50], highlighting the high specificity of this approach (Supplementary Data 5). The most predictive features included the number of pure ALT-TFs, the total number of ALT-TFs, the length of the breakpoint sequence, and the abundance of the TVRs TGAGGG and TTAGGG, which have been previously linked with ALT activity (Supplementary Fig. 14)[21,24]. Notably, we obtained a comparable sensitivity of detection even for non-ALT tumours (Supplementary Fig. 14D). This is consistent with studies reporting the coexistence of telomerase expression and ALT in the same cell populations in vitro[51,52] and in primary tumours[53–55].

Finally, as the genomic DNA in whole blood is of primarily hematopoietic origin, we sought to assess whether ALT-TFs are detected in cell-free DNA data from cancer patients but not in healthy individuals, which would thus be consistent with the somatic origin of ALT-TFs. To this aim, we performed WGS on 8 primary human glioblastomas (GBM) and matched whole blood and serum cell-free DNA samples, and then compared the estimated rates of ALT-TFs against published cell-free DNA WGS data from healthy controls[56] (Methods). We detected ALT-TFs in all the newly sequenced glioblastomas (Fig. 4D and Supplementary Table 2), which showed ALT-TF rates comparable to glioblastoma samples from PCAWG (Fig. 4E). The rates of ALT-TFs in the whole blood and serum cell-free DNA were overall lower as compared to the tumour samples, but were significantly higher than those estimated for cell-free DNA WGS data from healthy individuals[56] ($P = 5.9 \times 10^{-5}$, two-sided Wilcoxon test, Fig. 4E and Methods). As expected, the rate of chromosome 9 endogenous fusions, which we use as an internal control, were comparable across data sets and sample types ($P = 0.84$, two-sided Wilcoxon test, Fig. 4F), thus suggesting that the ALT-TFs that we detect are associated with the presence of cancer.

To evaluate the potential of ALT-TFs as a cancer biomarker for liquid biopsy analysis, we estimated the tumour fraction in cell-free DNA from glioblastoma patients and healthy controls using ichorCNA[57], and compared them against the ALT-TF rates. This analysis revealed a significant difference in the rate of ALT-Ts between glioblastoma patients and controls (Supplementary Fig. 15). By contrast, the tumour fraction values estimated using ichorCNA were not significantly different, indicating the added value of ALT-TF detection in plasma for liquid biopsy analysis.

Altogether, these results indicate that the detection of somatic ALT-TFs in blood represents a highly specific biomarker for liquid biopsy analysis.

## Discussion

Here, we report the discovery of a type of telomere fusions (ALT-TFs), which we link mechanistically with the activity of the ALT pathway. ALT-TFs permit the identification of the telomere maintenance mechanism of tumours in a quantitative manner using sequencing data, thus representing an alternative approach to current methods[24,58]. Using a non-invasive method to determine the telomere maintenance mechanism of tumours could facilitate the introduction of the ALT-status biomarker in clinical practice, which could inform prognosis and treatment selection[59].

Two features allow to distinguish ALT-TFs from the conventional TFs fusing two chromosomes: (i) the presence of outward fusions, and (ii) their localization to DNA fragments rather than chromosomal regions. The prevalence of outward over inward fusions could be attributed to localised telomere deprotection due to the release of sheltering complexes at the location of double-strand break, which is required for the formation of outward, but not inward, fusions. In addition, we found that microhomology at the fusion point is common in ALT-TFs. Microhomology results from the annealing of the self-complementary TA sequence, which is contained within canonical TTAGGG telomeric repeats. The relevance of the TA sequence in the formation of telomere fusions is consistent with previous studies indicating an increment of telomere fusions after artificially increasing the self-complementarity of telomere sequences[14]. Since telomere fusions can halt cell division, we infer that avoiding excessive self-complementarity in telomeric repeats would favour tumour expansion. Supporting this view, human tumours contain numerous TVRs that lack self-complementarity. Specifically, the most conspicuous TVRs found in human tumours (T**G**AGGG; T**C**AGGG; TT**G**GGG; TT**C**GGG)[21,32] destroy the self-complementary TA sequence. In addition, single TVRs are interspersed throughout canonical TTAGGG sequences in cells positive for ALT markers limiting the chances of self-complementary alignments[21]. Based on these observations, we hypothesize that telomere sequences that lack self-complementarity are strongly selected for in tumours in which telomere breakage (deprotection) is favoured, as is the case in ALT positive cells.

Previous work has shown that ALT-specific C-circles can be detected in the whole blood and plasma cell-free DNA from cancer patients[60]. Here, we show that ALT-TFs are detected in the blood of cancer patients, which can be used for liquid biopsy analysis with high sensitivity and high specificity. We obtain high sensitivity for tumour types for which early detection methods are severely lacking, such as pancreatic cancer, and tumours that are difficult to sample, such as pediatric brain tumours and glioblastomas. Notably, we report high sensitivity and specificity for predictive models trained on whole blood WGS data generated using standard protocols for germline genomic analysis. Although we cannot rule out the possibility that the ALT-TFs we detect in whole blood and serum cell-free DNA from cancer patients might be generated by non-neoplastic cells, our analysis of cell-free DNA WGS data from glioblastoma patients and healthy controls suggests that the ALT-TFs detected in blood are linked with tumorigenesis. Enrichment or amplification of DNA fragments containing ALT-TFs followed by sequencing or other readouts[61] could further improve sensitivity and scalability, thus complementing current liquid biopsy methods[62,63]. Overall, detection of ALT-TFs in blood and cell-free DNA represents a method for liquid biopsy analysis, with implications for early detection, patient stratification, disease monitoring and treatment selection[64].

## Methods
### Human subjects
The glioblastoma and blood samples used in this study were collected in the Neurosurgery Department at Centro Hospitalar Universitário Lisboa Norte (CHULN) and stored at Biobanco-iMM CAML (Lisbon Academic Medical Center, Lisbon, Portugal). Ethical approval was obtained from the Ethics Committee of CHULN (Ref. Nº 367/18). Written informed consent was obtained from all patients prior to study participation in accordance with the European and National Ethical Regulation (law 12/2005). Patients did not receive financial compensation for donating samples. Data collection was carried out by a health care professional and included age and self-reported sex information. Surgical samples, not needed for diagnostic purposes, were processed less than 1 h after surgery and included in the brain tumour collection of Biobanco-iMM CAML.

## Data sets

To characterize the landscape of somatic TFs in human cancers, we analysed WGS data from 2071 tumour-normal sample pairs from the PCAWG consortium[65], 306 cancer cell lines from the CCLE[25], 119 blood samples from the TARGET project, and 438 blood samples from the CPTAC consortium[66,67]. In addition, to assess the rate of TFs in non-cancer samples, we curated WGS data sets from 255 blood samples from the GTEx project[68], 304 blood control samples from control individuals from the Genetic Epidemiology of COPD project (COPD-Gene), which is part of the TOPMed Program, 2490 EBV-transformed cell lines from the 1000 Genomes Project[69], 99 blood samples from the KPGP project[49], and 279 non-neoplastic samples from the Panbody study (from individuals PD43851/PD42565, PD43850 and PD28690)[50]. Finally, to elucidate the underlying mechanisms of TF formation, we analysed WGS data from 117 retinal pigment epithelium (RPE-1) clones sequenced upon induction of telomere crisis[10] and 7 controls[9]. The list of data sets, sample IDs, and alignment details for each sample are provided in Supplementary Data 1.

## Processing of surgical tumour tissue and matched normal samples

Tumour tissue samples were divided into small fragments (~0.5 x 0.5 cm) in a sterile petri dish with a sterile scalpel. The fragments were snap-frozen and stored in liquid nitrogen. Blood samples were collected in EDTA anticoagulant tube (Sarstedt) and stored in an ultra-freezer at −80 °C. For serum separation, blood samples were also collected in clot activator tubes (Sarstedt), centrifuged at 2000g for 10 min at room temperature. Serum samples were then stored in an ultra-freezer at −80 °C. Whole-genome sequencing was performed on the Illumina NovaSeq6000 sequencing machine using S4 flow cells to generate 2 x 151 paired-end reads. Quality control for all steps was performed using a DNA Screentape on the Agilent TapeStation system.

## Detection of telomere fusions using short-read sequencing data

We developed TelFusDetector to detect TFs using sequencing data[20]. To identify regions containing fusion-like patterns in the human genome that could be misclassified as somatic, we first scanned the human reference genome (builds hg19, GRCh38 and T2T-CHM13) for patterns of TFs in windows of 500bp with an overlap of 250bp. As expected, this analysis yielded the relic of a telomere fusion in chromosome 2[23]. In addition, we discovered additional regions with forward and reverse telomere repeats in chromosome 9, which we term "chr9 endogenous fusion". The coordinates for the genomic regions in the reference genome builds hg19, GRCh38 and T2T-CHM13 containing telomeric repeats are provided in Supplementary Table 1.

To detect somatic TFs in sequencing data sets, we extracted aligned sequencing reads containing at least two consecutive TTAGGG and two consecutive CCCTAA telomere sequences using custom python scripts relying on the Pysam and regex modules (https://github.com/pysam-developers/pysam). We allowed for up to two mismatches in the telomeric repeats to consider TVRs[22]. Next, we extracted the mate for each read containing telomeric repeats, and filtered out duplicate reads, as well as supplementary and complementary alignments. Reads mapping to regions in the reference genome containing telomere fusion-like patterns were discounted as somatic.

For the detection of TFs in ALaP-Seq data we used an alternative pipeline that applied FastQC to control for the quality of reads, Cutadapt to remove adapter sequences, and in-house Perl scripts. First, we identified candidate reads containing putative fusions by searching for the presence of two consecutive circularly permuted TTAGGG motifs and two consecutive, circularly permuted CCCTAA motifs. Reads mapping to endogenous fusions identified in the mouse reference genome (mm10) were discarded. Finally, we classified the ALT-TFs by analysing the relative orientation of the circularly permuted TTAGGG and CCCTAA repeats and the length of the breakpoint sequence.

## Quality control of telomere fusion calls

To increase the specificity of the ALT-TF calls, we applied a set of filters to remove low-quality read pairs based on alignment quality information and the sequence context of the telomeric repeats. Specifically, we filtered out read pairs with candidate ALT-TFs if (i) at least one read in the pair contained more than one fusion breakpoint of the outward or inward type; (ii) the mate read length was greater than 60bp, did not contain telomeric repeats, and mapped with MAPQ >= 8 to a non-subtelomeric chromosomal region[21]; or (iii) at least one read in the pair mapped to an endogenous TF. The coordinates for telomere regions were downloaded from http://genome.ucsc.edu. Subtelomeric regions were defined as 100 Kbp regions from each chromosomal end. Read pairs supporting the same ALT-TF in each read, as defined by identity of the breakpoint sequence and type of fusion, were considered to originate from fragments with a short insert size and were thus collapsed into a single read. Those read pairs where both mates supported ALT-TFs in the same orientation but with distinct, non-reverse complement breakpoint sequences were discarded.

After applying these filtering criteria, we removed low-quality samples. Specifically, we discarded samples with <5 reads mapping to the chromosome 9 endogenous fusion, or if less than 90% of the reads mapping to this region did not show the expected breakpoint sequence TTAA, which corresponds to the 1% quantile estimated utilizing the PCAWG cohort. In addition, we removed samples with an unusually high rate of filtered reads. Specifically, we discarded the top 1% samples with the highest fraction of sequencing reads with candidate ALT-TFs filtered out in the previous steps.

## Classification of telomere fusions

We classified ALT-TFs into 3 categories based on the relative position of the telomeric sequences TTAGGG and CCCTAA. We classified ALT-TF characterized by 5′-TTAGGG-3′ telomeric repeats followed by 5′-CCCTAA-3′ as inward, and those with 5′-CCCTAA-3′ repeats followed by 5′-TTAGGG-3′ as outward. Finally, read pairs where a read in the pair supports an inward ALT-TF and the other an outward ALT-TF were classified as circular (in-out) events, as this pattern would be expected for ALT-TF formed through ligation and subsequent circularization of short terminal telomere fragments.

In addition, we classified ALT-TFs into pure or alternative based on the length and type of sequence at the breakpoint junction. Specifically, we extracted the sequence flanked by the telomeric repeats TTAGGG and CCCTAA allowing for one mismatch or indel in each of the repeats to account for telomere variant repeats and sequencing errors. ALT-TFs with breakpoint sequences in the set of all possible permutations of TTAGGG and CCCTAA were classified as pure, and those with breakpoint sequences longer than 12bp as alternative.

## Benchmarking of TelFusDetector using simulations

To evaluate the performance of TelFusDetector, we simulated tumours with telomere fusions and normal samples without at different sequencing depths. To this aim, we first created telomeric contigs with telomeric fusions by fusing in silico telomeric sequences extracted from the T2T-CHM13 assembly. Specifically, to simulate a telomere fusion we first extracted the 2500 terminal base pairs from a pair of chromosomes randomly selected. Next, we simulated a double-strand break at a random location within each 2500bp sequence and fused two of the four resulting fragments in either inward or outward orientation (each with a 0.5 probability). We simulated a total of 184 ALT-TFs (100 inward and 84 outward), and added them as new contigs to the T2T-CHM13 assembly. We next simulated paired-end reads for the entire genome including the canonical contigs and those containing telomere fusions using *wgsim*[70] (default parameters with base

error rate ~ 0.02, and read length of 150bp) to generate simulated sequencing runs spanning a dynamic range of sequencing coverage: 1x, 5x, 10x, 20x, 40x, 60x, and 100x. In addition, we simulated normal samples in the same manner using the T2T-CHM13 reference genome without the contigs containing telomere fusions. Finally, we aligned the sequencing reads against the Hg38 reference genome using BWA-MEM, as this combination of reference genome and aligner is the most frequently used in the data sets we have analysed in this study.

To estimate the sensitivity of TelFusDetector, we used the simulated tumour samples harbouring ALT-TFs, while we estimated precision through the analysis of the simulated normal samples, where no ALT-TFs should be detected. Given that the read length in most data sets analysed in this study is 150bp, and that TelFusDetector requires at least two TTAGGG and two CCCTAA repeats to make an ALT-TF call, we used simulated ALT-TFs for benchmarking purposes whose breakpoint sequence is equal or shorter than 126bp, which corresponds to the largest breakpoint sequence we can detect with sequencing reads of 150bp.

To simulate different levels of tumour cellularity, we mixed simulated sequencing reads for the samples with and without ALT-TFs at different fractions to reach a total sequencing depth of 60x.

### Estimation of the fragment size of DNA fragments containing ALT-TFs

To assess whether ALT-TFs originate from short DNA fragments, we compared the insert size distribution for read pairs containing ALT-TFs and read pairs mapping outside telomeric regions. For this analysis, we focused on ALaP-Seq paired-end libraries aligned against the mm10 assembly using bowtie2[71]. We estimated the size of the DNA fragments containing ALT-TFs in ALaP-Seq paired-end libraries by assembling reads containing ALT-TFs and their mates using the EMBOSS Merger[72]. Given the low complexity of telomere repeat-containing sequences, we only considered read pairs satisfying the following criteria: (i) the same ALT-TF was detected in both reads in the pair; and (ii) the overlapping region in the assembled sequence was at least 10bp long and with no mismatches, and at least 10% divergent from canonical telomeric repeats, including TVRs. As a baseline for comparison, we estimated the fragment length for read pairs in the same library not mapping to telomeric regions. Alignments were filtered to keep only paired and properly mapped reads. We only considered for further analysis read pairs with an estimated insert size shorter than 190bp given that we require an overlap of at least 10bp in the assembly of read pairs containing ATL-TFs.

### Identification of breakpoint sequences in TFs

To estimate the error rate in reads containing fusions, which limit their accurate identification, we analysed sequencing reads mapping to the chromosome 9 endogenous region in tumours and matched blood samples from PCAWG, as well as in blood samples from GTEx and TOPMed (Supplementary Fig. 1A-D). Because the chromosome 9 endogenous fusion is fixed in the population, only one breakpoint sequence should be detected in each sample (i.e., TTAA), and any additional breakpoint sequences would be the consequence of sequencing errors in the flanking TTAGGG or CCCTAA repeats. Using this approach, we estimated a base error rate at telomeric repeats of 0.74%, which is comparable to the expected sequencing error rate for Illumina data[73]. We detected at least two distinct breakpoint sequences (including TTAA) in 1725 of the 2163 samples (79.8%), indicating the relevance of accounting for sequencing errors for the accurate characterization of ALT-TF breakpoints. Using the reads mapping to the chr9 endogenous fusion, we developed a set of rules for error correction to account for mismatches and indels in the flanking telomeric repeats, which are available at https://github.com/cortes-ciriano-lab/TelFusDetector. This correction

step reduced the average number of distinct breakpoint sequences for the chr9 endogenous fusion to 1.07 and identified the breakpoint sequence TTAA only in 93.3% of cases (Supplementary Fig. 1B).

To assess the enrichment of microhomology tracts at the breakpoint sequence in pure ALT-TFs, we compared the observed against the expected number of occurrences of each breakpoint sequence based on all possible circular permutations of TTAGGG and CCCTAA sequences.

### Calculation of telomere fusion rates

To compare the number of ALT-TFs across samples aligned against difference builds of the human reference genome and sequenced using varying sequencing read length and depth, we computed a telomere ALT-TF rate for each sample as:

$$Telomere\,fusion\,rate = \frac{\frac{Num.reads}{with\,fusion}}{Purity} * \frac{1}{Coverage} \quad (1)$$

where

$$Coverage = \frac{Read\,length * library\,size}{3x\,10^9} \quad (2)$$

and *Library size* corresponds to the number of mapped reads in a sample after excluding duplicate reads and supplementary alignments. The purity values were obtained from PCAWG[65]. To calculate the rate of endogenous TFs in matched normal samples and cancer cell lines we used a tumour purity value of 1.

### Detection of telomere fusions using long-read sequencing

Due to the higher error rate of long-read as compared to Illumina sequencing, we required at least 15 TTAGGG and 15 CCCTAA telomeric repeats and at least 5 TTAGGGTTAGGG and 5 CCCTAACCTAA repeats to make a telomere fusion call in long reads. Sequencing reads mapping to non-subtelomeric regions with a mapping quality >= 8 and supplementary alignments were discarded. Fusion counts were then normalised for each sample by dividing the number of reads with ALT-TFs by the total number of reads in each WGS data set.

### Identification of genomic correlates of ALT-TF formation

To investigate the genomic correlates of ALT-TF formation, we first used a linear model to regress out the effect of the cancer type and tumour purity on the rate of ALT-TFs. We limited this analysis to tumour samples from the PCAWG project with a tumour purity of at least 30%. The residuals of the linear model were then used in a second linear model that incorporated age at diagnosis and sex as covariates, as well as other variables related to genome instability and telomere maintenance. Specifically, we encoded using categorical variables the presence of whole-genome doubling events[65],chromothripsis in at least one chromosome[74], truncating mutations in *TP53*, the number of SVs per sample, and chromosome instability signatures[75]. Tumour ploidy was encoded as a continuous variable. We also included as categorical covariates in the model the presence of activating mutations or SVs in *TERT* and the ALT status classification reported by de Nonneville et al.[24] or Sieverling et al.[21]. The telomere content normalized against the matched normal sample was included as a continuous variable[21]. Finally, due to the relationship between EBV transformation and ALT activity, we also included the presence of viral sequence insertion events in the tumour genome as a categorical variable[76]. In addition, for those samples with RNA-seq data available, we included the expression levels of *ATRX, TERF2, TERF1, TERT, DAXX* and *TERRA*. *TERRA* expression values were obtained from Sieverling et al.[21]. We performed backward stepwise model selection using the Akaike

information criterion to identify covariates associated with ALT-TF formation.

## Machine learning

We utilized the R package *RandomForest* to generate Random Forest (RF) models. We used default parameter values for model training, except for the number of trees, which was set to 100. To assess the robustness of the predictions, we trained a total of 100 RF models, each time assigning a different set of samples to the training and test sets. We selected as the optimal cut-off value the probability that maximized sensitivity and specificity across the 100 models. The predictive power of each covariate was estimated by computing the change in accuracy when excluding each covariate from the training data using the function *importance* from the *RandomForest* package.

## ALT prediction using ALT-TFs detected in tumour samples

We generated RF models for the prediction of the ALT status of tumours using the patterns of ALT-TFs detected in cancer samples as covariates. As a training set, we considered as ALT those tumours positive for the C-circle assay, or classified as ALT by both de Nonneville et al.[24] and Sieverling et al.[21] We then randomly assigned two-thirds of the ALT-positive samples and an equal number of non-ALT samples from cancer patients to the training set, and held out the rest for testing the model. We encoded the ALT-TFs detected in each sample using a set of 117 variables, which are listed in Supplementary Data 4.

## Cancer detection using ALT-TFs identified in whole blood samples

We trained RF models for the detection of cancer using the patterns of ALT-TFs detected in whole blood samples. For model training we focused on samples with at least 1 ALT-TF; individuals with no ALT-TFs detected in blood were considered to be cancer-free. We randomly assigned two-thirds of the GTEx, TOPMed and KPGP samples and an equal number of blood samples from cancer patients from the PCAWG, CPTAC and TARGET projects to the training set, and held out the remaining 1296 samples for testing the model. We encoded the ALT-TFs detected in each sample using the same set of 117 variables used to predict the ALT status of tumours (Supplementary Data 4). These include the fraction of each TVR in the reads supporting ALT-TFs and ALT-TF rates. Finally, the predictions of cancer status were obtained by selecting the most frequent prediction across 100 models, each trained on a random subset of the training data. As additional validation, we applied the obtained RF models to a set of 29 diverse histological structures from normal tissue samples from the Panbody study to investigate the specificity of our approach.

## Estimation of tumour fraction through liquid biopsy analysis

We used ichorCNA[57] (v0.3.2) to perform copy number analysis and estimate the fraction of tumour content in WGS data from serum cell-free DNA from glioblastoma patients and plasma cell-free DNA from healthy donors. In brief, wig files containing read count coverage information across the genome at 500Kb bins were generated using readCounter from the HMMcopy software suite (v0.99.0). Read count wig files were then analysed using ichorCNA and the following parameter values adjusted for samples with low tumour content: (1) the initial ploidy parameter was set to 2; (2) the non-tumour fraction initializer values were set to c(0.85, 0.9, 0.95, 0.99, 0.995, 0.999); (3) the maximum number of copy number states was reduced to 4; (4) no states were used to account for subclonal copy number events; (5) only autosomes were analysed and used for training. The tumour fraction solution with the highest log-likelihood was considered optimal.

## Statistics and reproducibility

Statistical analyses were performed in R (v.3.6). To assess statistical significance, two-tailed Wilcoxon's rank-sum test was used on continuous variables. Random forest modelling was performed using the R package randomForest (v4.6.14). ROC curves were computed using the R package pROC (v1.18.0). No statistical method was used to predetermine sample size.

## Reporting summary

Further information on research design is available in the Nature Portfolio Reporting Summary linked to this article.

## Data availability

The raw sequencing data from the PCAWG project is available through controlled access application to the International Cancer Genome Consortium Data Access Compliance Office (DACO; http://icgc.org/daco) for the ICGC portion, and to the TCGA Data Access Committee (DAC) via dbGaP for the TCGA portion (https://dbgap.ncbi.nlm.nih.gov/aa/wga.cgi?page=login; dbGaP Study Accession: phs000178.v1.p1). Additional information on accessing the PCAWG data, including processed data sets, can be found at https://docs.icgc.org/pcawg/data/. We used the following data sets from PCAWG, which are available at Synapse (https://www.synapse.org/) under the Synapse IDs syn10389158: clinical data from each patient, including tumour stage and vital status; syn1038916: harmonised tumour histopathology annotations using a standardised hierarchical ontology; and syn8272483: purity and ploidy values for each tumour sample. The raw sequencing data generated by the GTEx project is available through controlled access application via dbGAP (dbGaP Study Accession: phs000424.v8.p2). The raw sequencing data generated by the Genetic Epidemiology of COPD (COPDGene) project, which is part of TOPMed, is available through controlled access application via dbGAP (dbGaP Study Accession: phs000951.v4.p4). The raw sequencing data from the Korean Personal Genome Project (KPGP) were downloaded from SRA (study accession: PRJNA284338). Raw sequencing data from the CPTAC3 study is available through controlled access application to the NCI Data Access Committee (DAC) via dbGAP (dbGaP Study Accession: phs001287.v5.p4). WGS data from 124 retinal pigment epithelium (RPE) clones were downloaded from the European Nucleotide Archive database under primary accession number PRJEB23723[10] and European Genome-Phenome Archive (EGA, hosted by the EBI and the CRG) under the accession number EGAD00001001629[9]. The raw sequencing data from the Panbody project were downloaded from EGA (EGAD00001006641). PacBio and Illumina WGS data from the breast cancer cell line SK-BR-3 were downloaded from http://schatz-lab.org/publications/SKBR3/[26]. Nanopore and Illumina WGS data from the melanoma COLO829 cell line were downloaded from the European Nucleotide Archive (ENA) under the study accession PRJEB27698[27] PacBio WGS data from the colon cancer cell lines HCT116, KM12, SW620 and SW837 are available at the Gene Expression Omnibus (GEO) database under the accession number GSE149709[28]. AlaP-Seq and CHIRT-seq data are available at GEO under accession numbers GSE135563 and GSE79180, respectively. The raw sequencing data from the newly sequenced glioblastoma samples are available under controlled access at EGA under the accession number EGAD00001012101. Data access can be granted via the EGA for a defined time period after successful completion of a data access agreement provided by the WTSI CGP Data access committee (datasharing@sanger.ac.uk). The remaining data are available within the Article, Supplementary Information, Supplementary Datasets or Source Data files. Source data are provided with this paper.

## Code availability

The code to detect ATL-TFs using sequencing data is available at: https://github.com/cortes-ciriano-lab/TelFusDetector with https://zenodo.org/badge/latestdoi/676864666.

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

## Acknowledgements

F.M., C.S. and I.C.-C. thank EMBL and The Wellcome Trust for funding. Sequencing experiments were supported by Cancer Research UK award 30306 (I.C.-C.). I.F. was funded by grants from the Spanish Ministry of Science and Innovation (PID2019-110339RB-I00) and the Comunidad de Madrid (S2022/BMB-7245). The CNIC and CBM-SO are supported by the Ministerio de Ciencia, Innovación y Universidades and are Severo Ochoa Centers of Excellence (CEX2020-001041-S, CEX2021-001154-S). R.C., A.A. and C.C.F. acknowledge the support of Associação David Vaz, Bolsa João Lobo Antunes—GAPIC (iMM/FMUL) and Millennium bcp. All authors thank the computational resources provided by the European Bioinformatics Institute (EMBL-EBI). The authors acknowledge the patients who kindly provided the biological samples used for this research and the Biobanco-iMM CAML, which enabled the collection, processing, and storage of tumour and blood samples from glioblastoma patients. Some of the figures were created using BioRender.com.

## Author contributions

I.C.-C. and I.F. designed and supervised the study. I.C.-C. obtained funding for the experimental work. F.M. and M.J.G.R. performed analyses. F.M. generated the figures with input from M.J.G.R., I.C.-C., and I.F. F.M. and I.C.-C. implemented TelFusDetector with input from M.J.G.R. and I.F. C.M.S. contributed cell-free DNA analyses. F.M., I.C.-C., and I.F. wrote the manuscript with input from M.J.G.R. C.C.F., R.C., and A.A., collected and processed the tumour and blood samples. All authors read and approved the final manuscript.

## Funding

## Competing interests

F.M., M.J.G.R., I.C.-C. and I.F. have filed a patent application related to the discoveries disclosed in this manuscript [WO2023118606—DETECTION OF TELOMERE FUSION EVENTS; https://patentscope.wipo.int/search/en/detail.jsf?docId=WO2023118606&_cid=P20-LJGXTN-21697-1]. The remaining authors declare no competing interests.
