## [Peer Review File · Nature Communications]

REVIEWER COMMENTS

Reviewer #1 (Remarks to the Author): expertise in telomere fusion biology and ALT signalling

In this manuscript, Muyas et al., have analyzed telomere fusion fragments from whole genome sequencing data and found that telomere fusions are associated with the pathway involved in Alternative Lengthening of Telomeres (ALT). The telomere fusion fragments are characterized as inward and outward fusions termed ALT-associated telomere fusions (ALT-TFs). The authors show that ALT-TFs are found at TERRA binding sites and ALT-associated PML bodies in TERRA-CHIRT-seq and ALaP-seq data sets. They demonstrate that the breakpoint sequences within ALT-TFs contain microhomology, which may promote homology pairing and ligation of telomere fusions. Importantly, ALT-TFs can be detected in the blood of patients with cancers. Overall, their findings are interesting and may provide a new method for detecting cancers using blood samples or identifying ALT cancers by whole genome sequencing from cancer tissues. There are some concerns that need to be addressed before publication.

Major concerns:

1. Can ALT-TFs be a biomarker for ALT cancer using human blood samples? The data only presented the enrichment of ALT-TFs in various cancers but did not have a comparison of ALT and non-ALT cancers from blood samples.
2. Several studies (Hartilieb et al., 2021, Nat Comm; Lee and Pickett et al., 2018, Nucleic Acids Research) indicate that telomere content can be used to determine ALT activity in cancers. However, Figure 2A shows that the telomere content is not positively associated with inward or outward telomere fusion fragments. Can the authors compare which one (telomere content vs telomere fusions) is better for detecting ALT cancer?
3. Please provide the statistical analysis for Figure 4E. This is a very critical experiment stating that ALT-TFs could be biomarkers for cancer detection.
4. Telomere fusions and cell death can be induced by chemotherapy drugs that damage genomic DNA. Could the increase of TFs in blood samples be due to the treatment of chemotherapy drugs in cancer samples?

Minor comments:

1. There is no description for Figure 4F in the main text.
2. Please provide the figure legends in detail for Supplementary tables. For example, what are Sq, F values, Y_variable, and X_Variable in Supplementary table 3?
3. Please describe the method to determine TERRA levels from RNA-seq data. What are the data sets?

4. In Supplementary Figure 5B, please explain the features such as “pure” and “telomeric”.

5. Page 4: Please rephrase the subtitle “ALT-TFs bind to TERRA and localize to APBs”.

ALT-TFs are genomic fragments that could be associated with RNA and protein complexes.

The subtitle sounds like ALT-TFs are mobile and can bind to TERRA and travel to APBs. The subtitle could be “ALT-TFs are found in TERRA-binding sites and APBs.” Since mESCs don’t have high ALT activity, the presence of ALT-TFs in these cells should be discussed.

Reviewer #2 (Remarks to the Author): expertise in telomere fusions

Muyas et al. identify telomere fusions (TF) from whole genome sequencing (WGS) data of cancer and non-cancer samples. The team identifies not only the inward fusions as expected when two chromosomal ends join but also outward fusions. In a series of bioinformatics analyses, the study connects the TFs with alternative lengthening of telomeres (ALT), provides evidence that the TFs are small DNA fragments localized to ALT-associated promyelocytic leukaemia nuclear bodies (APBs), that microhomology is likely involved in the creation of TFs, develop a model how TFs emerge and define TFs as a biomarker for cancer in WGS data of patient’s blood.

The study is quite interesting with significant novelty which should be of interest to the field. I have the following comments which should be addressed.

Major points

1) Is it possible to derive from the data information on the origin of the telomere sequences, i.e. do the reads that contain fusions contain any sequence that allows to assign a specific chromosome and if so, is there any bias to specific chromosomes?

2) Could the authors speculate why outward TFs are more frequent compared to inward?

3) Page 7 related to Fig. 4E: “... The rates of ALT-TFs in the whole blood and serum cell-free DNA were overall lower as compared to the tumour samples, but were significantly higher than those estimated for whole blood and cell-free DNA WGS data from healthy individuals...”. Please provide statistical test and level of significance for this.

4) Why have samples without detected ALT-TFs not been included in the model training for the cancer detection? There should be significantly more such cases among non-cancer samples not being harmful for model training?

5) According to Fig. 3A/B it seems possible to me that inward TFs might not be driven by microhomology. Has the statistical test been corrected for the larger number of possible break points (7 and 4) that make the generation of TTAA and TAA/TTA fusion points more likely? In principle, this is also true of outward TFs but here is the separation between microhomology /no-homology more obvious.

Minor points

1) Supp Fig. 1B: color coding of the legend does not match figure.

2) How are the breakpoint sequences in Supp Table 5 defined? All sequences that are directly flanked by TTAGGG and/or CCCTAA that are not the canonical telomere sequences (accepting one mismatch)? Following that idea, I would have put in bold more than shown in "...outward ALT-TFs, the breakpoint sequence 5'...CCCTAACCTAGGGTTAGGG...3' was..." (middle of page 5).

3) Page 6, 3rd. paragraph: "...and remained comparable across cancer stages (Supplementary Fig. 9B-D)." should refer to Supp Fig. 10D.

4) Which samples is Supp Fig. 12D based on?

5) Not mandatory but out of curiosity: could the authors test whether the formation of TF does correlate with age in the non-cancer samples? If TF-formation is solely based on ALT, I do not expect a correlation with age.

6) Fig. 2J: APEX-PMLca and APEX-NLS should be explained.

7) Very minor: Fig. 3A Outward legend, 3rd from top seem to require also CCCAGGG?

Axel Hillmer

Reviewer #3 (Remarks to the Author): expertise in bioinformatics and gene fusions

This paper developed a computational tool called TelFusDetector for telomere fusion (TF) detection and analyzed whole genome sequencing data from PCAWG using TelFusDetector. The paper potentially has a high impact. It provided a first pan-cancer landscape of TFs, identified a novel TF pattern that might be closely related with the active lengthening of telomeres (ALT), and showed that TFs might serve as an important biomarker for cancer detection with liquid biopsy. Overall, the paper provides valuable resources and insights for following TF studies. I have the following concerns.

Major concerns:

1. Sensitivity and precision of TelFusDetector should be evaluated (e.g. by simulation).
2. The various TF-rate across cancers should be discussed in more details. For example, some cancer types are enriched with TF, like CNS tumors and sarcoma. Is this aligned with previous results in the field? What is the relationship between TF-rate and SV occurrence rate and mutations such as TP53?
3. More analyses on the association of the TF rate with other features relevant to chromosome instability, such as chromothripsis, breakage-fusion-bridge and the number of different patterns of SV (PMID: 32025012), should be analyzed. Some telomere-relevant features like TERT expression and TRF1/2 expression should also be included.
4. When calculating the TF rate, the authors divided the number of fusion reads by tumor purity. Since the estimation of tumor purity, either by histological review or by computational estimation, is not very accurate, such normalization is too dependent on purity and may lead to unexpected bias. I would suggest the authors to:
 - a) Show the correlations of TF rates with tumor purity before and after the purity correction to demonstrate the necessity and effect of the normalization.
 - b) First categorize the samples to different groups based on their purity (e.g., high, middle, low) and then repeat analyses in Fig. 2 using the raw TF rate without purity correction.
5. Another question about purity. When the authors analyzed the genomic correlates of ALT-TF, they used “linear model to regress out the effect of the cancer type and tumour purity on the rate of ALT-TFs”. However, if I understand correctly, the purity has already been considered when calculating the “rate of ALT-TFs”. Why is it regressed again in the model?
6. The authors successfully applied this method to liquid biopsy. However, as a biomarker for cancer detection, TFs should be compared with other biomarkers that can be inferred from ctDNA, such as fragmentome (PMID: 31142840, 34417454), presence of oncogenic mutations, etc.
7. I would recommend the authors to add more explanations about the TFs identified in normal tissues and blood cells. For example, in Fig. 4D (PD54225), there are significantly more TFs in whole blood than serum, indicating they were derived from normal blood cell. Is this biologically meaningful or only technical noise? Is there any way to remove them and retain only somatic TFs?

Minor concerns:

1. Figure 1B. If I understand correctly, each bar in the plot corresponds to the TF rate of one patient and the bars are ordered decreasingly by the TF rates. This should be clarified in the paper.
2. Definitions of some terminologies should be clarified for easier understanding, such as telomere content, TERTp as in Figure 2 and C-circle.

3. Figure 2, colors in many boxplots and barplots are hardly visible. Please consider replot (maybe in log scale)?

4. The genes used for evaluating ALT activity should be given in a supplementary table.

Page 4, “when the TERRA-AS not treated with RNase H or the TERRA sense transcript (TERRA-S) were used” should be “when the TERRA-AS was not treated with RNase H or the TERRA sense transcript (TERRA-S) was used”?

Response to Reviewers

We thank the Reviewers for their careful reading of our manuscript and their constructive comments. We think that the quality of the manuscript has been improved significantly as we modified the paper based on their comments. For example, we have now performed comprehensive simulations to assess the performance of TelFusDetector using the T2T-CHM13 genome assembly, and have now expanded on the analysis on the effect of tumour purity on the estimation of telomere fusion rates, among other analyses. In addition, we have now re-engineered the code of TelFusDetector to make it more user-friendly and efficient. Specifically, TelFusDetector can now be run using a single python command instead of a series of python, bash and R scripts, as was the case in the previous version (<https://github.com/cortes-ciriano-lab/TelFusDetector>). We summarise below the key changes and analyses that we have incorporated in response to the Reviewer's comments, followed by more detailed point-by-point responses.

General comments

Reviewer 1 asked for further analysis of the effect of chemotherapy treatment on the generation of ALT-TFs. Reviewer 2 asked whether we could identify which chromosomes are involved in the telomere fusions detected. Finally, Reviewer 3 asked to provide further justification on the effect of purity correction on the estimated telomere fusion rates, and to expand on the ctDNA analysis to compare the detection of ALT-TFs against other approaches to identify cancer through liquid biopsy analysis. We agree that these are important issues.

- Effect of chemotherapy on ALT-TF rates.** In this study, we have analysed WGS data from primary tumours from PCAWG, which encompasses data from The Cancer Genome Atlas (TCGA) and multiple ICGC projects. We note that TCGA focused on untreated, primary tumours, as indicated here: <https://www.cancer.gov/ccg/research/genome-sequencing/tcga/studied-cancers>. As can be seen in the Figure below, most of the tumours in PCAWG were treatment naive. In fact, only a subset of samples (77/2071, 3.7% of tumours; and 59/1604, 3.7% of whole blood samples) were from patients who had received chemotherapy, combined chemotherapy and radiation or other unspecified therapy.

To assess whether chemotherapy or other therapies might cause ALT-TFs, we have compared the rate of fusions across groups. For this analysis, we have focused on liver hepatocellular carcinoma (Liver-HCC) and ovarian adenocarcinoma (Ovary-AdenoCA) because these cancer types had the most balanced distribution of treated vs untreated tumours, and at least 5 tumours with treatment. As it can be seen in the Figure below, we did not find a significant difference in the rate of ALT-TFs between treatment-naive samples and those from individuals who received treatment before resection or biopsy.

Therefore, the rates of ALT-TFs we present in this study correspond in their vast majority to treatment-naive tumours, and in the small set of treated tumours in PCAWG, the fusions that we detect are not the consequence of chemotherapy treatment. We have now clarified this point in the main text.

- Identification of chromosomes involved in the observed fusions.** Human subtelomeres exhibit remarkable sequence variability and display a pronounced diversity among individuals (PMID: 18466090), which makes a formidable challenge to accurately map reads to distinct subtelomeric chromosomes. Moreover, situated at the boundary of telomeric and subtelomeric DNA resides a Telomere-associated repeat (TAR1) region, present in approximately 85% of all telomeres (<https://doi.org/10.1101/2022.05.09.491186>). Therefore, utilising short reads alone proves inadequate for discerning the specific chromosomes involved in the detected telomere fusions. Nevertheless, we have performed simulations to assess whether it is possible to identify the chromosomes involved in telomere fusions using short-read sequencing. Specifically, we have simulated telomere fusions of both the inward and outward type using the T2T-CHM13 genome assembly. Then, we have simulated sequencing reads of the same length as the read length in the data sets analysed encompassing the telomere fusions. Next, we mapped the simulated sequencing reads to the T2T-CHM13 genome using BWA-MEM. For this analysis, we have focused on those simulated telomere fusions in which at least one of the simulated reads originates from the subtelomeric region, which could identify the chromosome involved if aligning such reads to subtelomeric regions was reliable. We can assess whether the mapping to subtelomeric regions is reliable as we know which two chromosomes are involved in the simulated fusions. As can be seen below, the correct chromosome involved in the fusion can only be detected in a minority of cases (2.7%), as the mapping quality of the reads encompassing subtelomeric regions is low and in most cases unreliable (that is, the aligner maps the read to the subtelomeric region of

an incorrect chromosome). Overall, this analysis shows that we cannot confidently identify the chromosomes involved in the ALT-TFs detected. To map the per-chromosome rate of ALT-TFs long-read sequencing data from tumours is required, which is however not yet available at scale to perform such analysis.

Analysis of the mapping quality of reads originating from simulated ALT-TFs.

- Effect of tumour purity on the estimation of ALT-TF rates.** We have now studied the effect of tumour purity on the estimated rate of ALT-TFs using simulations. Specifically, we generated a version of the T2T-CHM13 genome assembly containing simulated telomere fusions and generated sequencing reads from this assembly. The telomere fusions were simulated by concatenating telomere sequences from multiple chromosomes - the breakpoint positions in the telomere sequences were selected randomly. To generate sequencing reads without somatic telomere fusions, we also simulated sequencing reads from the T2T-CHM13 genome assembly. Next, to simulate different levels of tumour cellularity, we mixed sequencing reads from both assemblies at different fractions to reach a total sequencing depth of 60x, which is comparable to the average sequencing depth in PCAWG samples. We analysed these simulated samples to estimate the ALT-TF rate. As can be seen below (right panel), the raw estimated ALT-TF rate is strongly affected by the level of tumour cellularity. However, after correcting for tumour purity, the rate is comparable across a dynamic range of tumour purities, which represent the range of purities observed in cancer samples. Therefore, these analyses indicate that correcting for tumour purity as we have performed in our study is essential to compare ALT-TF rates across samples of variable tumour cellularity.

(Left) ALT-TF rate estimated using simulated samples across increasingly higher values of simulated tumour cellularity (or purity) and without correcting for tumour purity. (Right) ALT-TF rates estimated for the same samples shown on the Left panel after correcting for purity. Note that the ALT-TF rate is the same in all samples, which is only correctly estimated after correcting for purity.

To further assess whether the purity correction we have used has an influence on the genomic correlates of ALT-TF formation that we report, we repeated the analysis using either purity-corrected ALT-TFs and raw ALT-TF rates without purity correction. In both analyses, we found that both inward and outward ALT-TFs are associated with the same covariates (see figure below), highlighting that the genomic correlates of ALT-TF formation that we report are strong and not artefacts of the purity correction step.

Coefficient values estimated using linear regression analysis and variable selection for the covariates with the significant positive and negative association with TF rates. The results on the left show the estimates using the purity-corrected ALT-TF rates, and the results on the right show the estimates obtained using the raw (without purity correction) ALT-TF rates.

Importantly, we note that the new release of TelFusDetector permits the user to decide whether to use purity information to correct the ALT-TF rates, thus increasing the versatility and applicability of our tool.

- Comparison of ctDNA analysis methods with ALT-TF detection.** We have now performed copy number analysis of the cell-free DNA samples from patients with human glioblastomas that we sequenced for this study using ichorCNA, which is the reference tool for this type of analysis in the field (PMID: 29109393). We applied ichorCNA to the serum samples from glioblastoma patients and control plasma samples and compared the results with the ALT-TF rates obtained with TelFusDetector.

This analysis has revealed no significant differences ($P > 0.05$; two-sided Wilcoxon test) in terms of tumour fraction estimated through copy number analysis between control plasma samples and the glioblastoma patients (see Figure below). These results are expected given the low volumes of circulating tumour DNA in plasma from glioblastoma patients, which limits the detection of copy number aberrations. However, the rate of ALT-TFs was different between plasma controls and glioblastoma patients ($P = 5.9e-5$ two-sided Wilcoxon test). In addition, the rate of chromosome 9 endogenous fusions, which we use as a control, was comparable between controls and glioblastoma patients. These results thus indicate a higher sensitivity for ALT-TFs to detect the presence of cancer as compared to copy number analysis.

(A) Comparison of ALT-TF rates detected in cell-free DNA from serum from glioblastoma (GBM) patients and cell-free DNA from control plasma samples. (B) Rate of chromosome 9 endogenous fusions in serum from GBM patients and plasma controls. (C) Comparison of the tumour fraction estimated using ichorCNA for the serum samples from GBM patients and plasma controls. The red bar represents the median. P values were computed using two-sided Wilcoxon tests.

Point-by-Point Responses

Reviewer #1 (Remarks to the Author): expertise in telomere fusion biology and ALT signalling

In this manuscript, Muyas et al., have analyzed telomere fusion fragments from whole genome sequencing data and found that telomere fusions are associated with the pathway involved in Alternative Lengthening of Telomeres (ALT). The telomere fusion fragments are characterized as inward and outward fusions termed ALT-associated telomere fusions (ALT-TFs). The authors show that ALT-TFs are found at TERRA binding sites and ALT-associated PML bodies in TERRA-CHIRT-seq and ALaP-seq data sets. They demonstrate that the breakpoint sequences within ALT-TFs contain microhomology, which may promote homology pairing and ligation of telomere fusions. Importantly, ALT-TFs can be detected in the blood of patients with cancers. Overall, their findings are interesting and may provide a new method for detecting cancers using blood samples or identifying ALT cancers by whole genome sequencing from cancer tissues. There are some concerns that need to be addressed before publication.

We thank the Reviewer for these positive comments.

Major concerns:

1. Can ALT-TFs be a biomarker for ALT cancer using human blood samples? The data only presented the enrichment of ALT-TFs in various cancers but did not have a comparison of ALT and non-ALT cancers from blood samples.

We thank the reviewer for bringing up this point. The differences in the rate of ALT-TFs in whole blood between ALT and non-ALT tumours varies across cancer types (Figure below). We note that the number of ALT-TFs in blood can be influenced by many factors, including tumour stage, the level of DNA shedding, and the amount of circulating and necrotic cells, as well as parameters related to the biology of cancer cells beyond the ALT pathway.

Comparison of ALT-TF rates estimated using whole-blood samples from patients with ALT and non-ALT tumours.

To fully understand the factors determining the differences in ctDNA shedding between ALT and non-ALT tumours, a larger number of WGS data sets from cell-free DNA samples would be needed, which are not readily available in the public domain currently. We believe that these further analyses are out of the scope of this work and leave them for future studies.

2. Several studies (Hartilieb et al., 2021, Nat Comm; Lee and Pickett et al., 2018, Nucleic Acids Research) indicate that telomere content can be used to determine ALT activity in cancers. However, Figure 2A shows that the telomere content is not positively associated with inward or outward telomere fusion fragments. Can the authors compare which one (telomere content vs telomere fusions) is better for detecting ALT cancer?

We thank the Reviewer for this excellent suggestion. To address this point, we have trained Random Forest models to predict ALT status based on: (1) 117 features of the telomere fusions, (2) one covariate encoding telomere content information, and (3) both telomere fusion features and telomere content information.

As shown in the Figure below, the model using only telomere fusion information yielded more accurate ALT predictions compared to those based only on telomere content, with AUC values of 0.93 and 0.88, respectively. In addition, incorporating both telomere fusion and telomere content information marginally enhanced the ALT prediction accuracy compared to using telomere fusion alone (AUC values of 0.94 versus 0.93). To evaluate the importance of telomere content information and the features of ALT-TF for the prediction of ALT status, we assessed the predictive power assigned to each covariate ("Mean decrease accuracy") by the Random Forest model. For this analysis, we used the Random Forest most trained on both telomere fusion and telomere content information (referred to as number 3 above). Telomere content was ranked top, followed by the rate of outward ALT-TF and other features of ALT-TFs (see Figure below). Therefore, these results indicate that both telomere content and the features of ALT-TFs are strong predictors of ALT, and that the combination of both sources of information provides higher predictive power to identify ALT tumours.

Performance of Random Forest (RF) models trained to predict ALT status based on the features of the ALT-TFs detected in tumours from the PCAWG data set. (A) AUC-ROC curve obtained for the RF model trained using features of the ALT-TFs as covariates; (B) AUC-ROC curve obtained for the RF model trained using telomere content information; (C) AUC-ROC curve obtained for the RF model trained using the combination of ALT-TF features and telomere content information. (D) Mean decrease in accuracy for the covariates included in the RF model trained on both telomere content information and ALT-TF features.

3. Please provide the statistical analysis for Figure 4E. This is a very critical experiment stating that ALT-TFs could be biomarkers for cancer detection.

We have now provided statistical analysis for Figure 4E.

4. Telomere fusions and cell death can be induced by chemotherapy drugs that damage genomic DNA. Could the increase of TFs in blood samples be due to the treatment of chemotherapy drugs in cancer samples?

Please see our response in General Comments.

Minor comments:

1. There is no description for Figure 4F in the main text.

We now cite Figure 4F in the main text.

2. Please provide the figure legends in detail for Supplementary tables. For example, what are Sq, F values, Y_variable, and X_Variable in Supplementary table 3?

Thanks to the reviewer for pointing out the lack of explanation in this supplementary table. For easier understanding, we have now included this table as *Supplementary Figure 5*.

3. Please describe the method to determine TERRA levels from RNA-seq data. What are the data sets?

TERRA expression levels for the PCAWG data set were obtained from Sieverling et al. 2020 (PMID: 32024817). To do so, the authors ran TelomereHunter on the RNA-seq bam files to count the number of reads containing the most common telomeric repeats and normalised this count by the total number of reads in each sample. We have now clarified how TERRA levels were obtained in the Methods section.

4. In Supplementary Figure 5B, please explain the features such as “pure” and “telomeric”.

We have now added these definitions in the caption of Supplementary Figure 5B.

Pure refers to the rate of ALT-TFs with breakpoint sequences in the set of all possible permutations of TTAGGG and CCCTAA, whereas *Telomeric* refers to the rate of all ALT-TFs excluding those whose supporting reads map to subtelomeric regions with mapping quality (MAPQ) > 8.

5. Page 4: Please rephrase the subtitle” ALT-TFs bind to TERRA and localize to APBs”. ALT-TFs are genomic fragments that could be associated with RNA and protein complexes. The subtitle sounds like ALT-TFs are mobile and can bind to TERRA and travel to APBs. The subtitle could be” ALT-TFs are found in TERRA-binding sites and APBs.” Since mESCs don’t have high ALT activity, the presence of ALT-TFs in these cells should be discussed.

We thank the reviewer for noting this. We have changed the manuscript’s subtitle to: “ALT-TFs are found in TERRA-binding sites and APBs” as the reviewer suggested.

We have also discussed the presence of ALT-TF in mESCs adding the next sentence to the manuscript: “Furthermore, the above findings indicate the presence of ALT-TF within mESCs, a cell type that not only exhibits telomere activity but also encompasses ALT features, exemplified by the lengthening of their telomeres through an homologous recombination mechanism mediated by Zscan4 (new ref. 36)”.

Reviewer #2 (Remarks to the Author): expertise in telomere fusions

Muyas et al. identify telomere fusions (TF) from whole genome sequencing (WGS) data of cancer and non-cancer samples. The team identifies not only the inward fusions as expected when two chromosomal ends join but also outward fusions. In a series of bioinformatics analyses, the study connects the TFs with alternative lengthening of telomeres (ALT), provides evidence that the TFs are small DNA fragments localized to ALT-associated promyelocytic leukaemia nuclear bodies (APBs), that microhomology is likely involved in the creation of TFs, develop a model how TFs emerge and define TFs as a biomarker for cancer in WGS data of patient's blood. The study is quite interesting with significant novelty which should be of interest to the field. I have the following comments which should be addressed.

We thank the Reviewer for these positive comments.

Major points

1) Is it possible to derive from the data information on the origin of the telomere sequences, i.e. do the reads that contain fusions contain any sequence that allows to assign a specific chromosome and if so, is there any bias to specific chromosomes?

Please see our response in General Comments above.

2) Could the authors speculate why outward TFs are more frequent compared to inward?

The generation of outward TFs implies the fusion between telomeric fragments generated by double strand breaks (DSB). These DSB likely concur with the release of the sheltering complex at the location of the break, provoking a local telomere deprotection and subsequently facilitating telomere fusion, given that deprotected telomeres tend to fuse.

Consequently, the prevalence of outward TFs over inward TFs could be attributed to localised telomere deprotection due to DSB. This disparity arises due to the fact that outward TFs require a DSB, unlike inward (end-to-end) TFs, where the presence of a DSB is not imperative for their formation. We have added this speculation in the manuscript's discussion section.

3) Page 7 related to Fig. 4E: "... The rates of ALT-TFs in the whole blood and serum cell-free DNA were overall lower as compared to the tumour samples, but were significantly higher than those estimated for whole blood and cell-free DNA WGS data from healthy individuals...". Please provide statistical test and level of significance for this.

We have now provided the results of the statistical analysis for Figure 4E in the main text.

4) Why have samples without detected ALT-TFs not been included in the model training for the cancer detection? There should be significantly more such cases among non-cancer samples not being harmful for model training?

The main goal of the Random Forest model trained for cancer detection was to understand the pattern of telomere fusions that could distinguish between cancer and cancer-free individuals using features related to the rates and properties of ALT-TFs detected in whole blood. Samples with no ALT-TFs detected are thus non-informative for the model in that the associated feature vector consists entirely of 0s. We note that when we applied the trained model to samples with no ALT-TFs (that is, whose associated vector covariates consist of 0s only) were predicted as cancer-free.

5) According to Fig. 3A/B it seems possible to me that inward TFs might not be driven by microhomology. Has the statistical test been corrected for the larger number of possible break points (7 and 4) that make the generation of TTAA and TAA/TTA fusion points more likely? In principle, this is also true of outward TFs but here is the separation between microhomology /no-homology more obvious.

We recognize that the explanation of this analysis in the Methods section might have been misleading. We have now clarified this section. Specifically, to assess the enrichment of microhomology tracts at the breakpoint sequence in pure ALT-TFs, we compared the observed number of occurrences of each possible sequence against all possible circular permutations of TTAGGG and CCCTAA. For instance, two circular permutations (i.e., without microhomology) can result in the breakpoint sequence GGG/CCC. In addition, two microhomology-mediated events can also result in the breakpoint sequence GGG/CCC. To compute whether there is an excess of ALT-TFs with the breakpoint sequence GGG/CCC over the expected rate, we consider the expected fraction to be two (corresponding to the two circular permutations that can lead to a breakpoint sequence of GGG/CCC) divided by the total number of possible circular permutations. Based on this expected fraction, we can then compute whether there is an enrichment of ALT-TFs with this breakpoint sequence. In this case, we attribute the enrichment that we observe to the two extra microhomology-mediated events that can result in the breakpoint sequence GGG/CCC.

Minor points

1) Supp Fig. 1B: color coding of the legend does not match figure.

We have now corrected the legend of Supplementary Figure 1B.

2) How are the breakpoint sequences in Supp Table 5 defined? All sequences that are directly flanked by TTAGGG and/or CCCTAA that are not the canonical telomere sequences (accepting one mismatch)? Following that idea, I would have put in bold more than shown in "...outward ALT-TFs, the breakpoint sequence 5'...CCCTAACCTAGGGTTAGGG...3' was..." (middle of page 5).

We thank the reviewer for pointing out this issue. The Reviewer is indeed correct. We have now corrected the sentence on page 5 to:

"In the case of outward ALT-TFs, the breakpoint sequence 5'...CCCTAAC**CTAGGG**TTAGGG...3' was the most abundant (22% of pure ALT-TFs) followed by 5'...CCCTAAC**CTT**AGGG...3' (16%)"

3) Page 6, 3rd. paragraph: “...and remained comparable across cancer stages (Supplementary Fig. 9B-D).” should refer to Supp Fig. 10D.

We have corrected the reference to Supplementary Figure 10D (now Supplementary Figure 12D in the current version of the manuscript).

4) Which samples is Supp Fig. 12D bases on?

We have now included in the caption of the Figure the samples included in Supplementary Figure 12D (now Supplementary Figure 14D in the current version of the manuscript).

5) Not mandatory but out of curiosity: could the authors test whether the formation of TF does correlate with age in the non-cancer samples? If TF-formation is solely based on ALT, I do not expect a correlation with age.

We agree with the Reviewer that this is an interesting point. We evaluated the correlation of the rates of both inward and outward ALT-TFs detected in the blood of the healthy donors with age. Our analysis did not reveal a significant correlation between ALT-TF rates and age ($P > 0.05$).

Distribution of the rate of ALT-TFs in whole blood from healthy donors as a function of age.

6) Fig. 2J: APEX-PMLca and APEX-NLS should be explained.

We have now included an explanation in the legend of Figure 2.

7) Very minor: Fig. 3A Outward legend, 3rd from top seem to require also CCCAGGG?

We have now added CCCAGGG to the legend of Figure 3A.

Axel Hillmer

Reviewer #3 (Remarks to the Author): expertise in bioinformatics and gene fusions

This paper developed a computational tool called TelFusDetector for telomere fusion (TF) detection and analyzed whole genome sequencing data from PCAWG using TelFusDetector. The paper potentially has a high impact. It provided a first pan-cancer landscape of TFs, identified a novel TF pattern that might be closely related with the active lengthening of telomeres (ALT), and showed that TFs might serve as an important biomarker for cancer detection with liquid biopsy. Overall, the paper provides valuable resources and insights for following TF studies. I have the following concerns.

We thank the Reviewer for these positive and encouraging comments.

Major concerns:**1. Sensitivity and precision of TelFusDetector should be evaluated (e.g. by simulation).**

We thank the Reviewer for this excellent suggestion. To evaluate the performance of TelFusDetector, we simulated tumours with telomere fusions and normal samples without at different sequencing depths. To this aim, we first created telomeric contigs with telomeric fusions by fusing *in silico* the telomeric sequences from the T2T-CHM13 assembly. Specifically, to simulate a telomere fusion we first extracted the 2500 terminal base pairs from a pair of chromosomes randomly selected. Next, we simulated a double strand break at a random location within each 2500bp sequence, and fused two of the four resulting fragments in either inward or outward orientation (each with a 0.5 probability) in cases when at least two fragments were at least 500bp long. We simulated a total of 184 telomere fusions (100 inward and 84 outward), and considered each of them as a contig, and added them as new contigs to the T2T-CHM13 assembly. We next simulated paired-end reads for the entire genome including the canonical contigs and those containing telomere fusions using *wgsim* (default parameters with base error rate ~ 0.02, and read length of 150bp) to generate simulated samples spanning a dynamic range of read depth: 1x, 5x, 10x, 20x, 40x, 60x, and 100x. In addition, we simulated normal samples in the same manner using the T2T-CHM13 reference genome without the contigs containing telomere fusions. Finally, we aligned the sequencing reads against the Hg38 reference genome using BWA-MEM, as this combination of reference genome and aligner is the most frequently used in the data sets we have analysed in this study.

To estimate the sensitivity of TelFusDetector, we used the simulated tumour samples harbouring ALT-TFs, while we estimated precision through the analysis of the simulated normal samples, where no ALT-TFs should be detected. Given that the read length in most data sets analysed in this study is 150bp, and that TelFusDetector requires at least two TTAGGG and two CCCTAA repeats to make an ALT-TF call, we used simulated ALT-TFs for benchmarking whose breakpoint sequence is equal or shorter than 126bp, as this is the largest breakpoint sequence we can detect with sequencing reads of 150bp. However, we note that among ~20,000 simulated ALT-TFs using the T2T-CHM13 assembly, only 5% had a breakpoint sequence longer than 126pb (see Figure below), indicating that we can call the majority of ALT-TFs using short reads.

Distribution of breakpoint lengths for 20,000 simulated ALT-TFs using the T2T-CHM13 assembly.

As shown in the Figure below (left), sensitivity increased with coverage. However, sensitivity was 0.91 (inward ALT-TFs) and 0.86 (outward ALT-TFs) for 5x. The sensitivity of TelFusDetector was 0.90-0.95 (for inward and outward ALT-TFs) for sequencing depths of 40x and 60x, respectively, which is the average coverage of PCAWG samples (PMID: 36697834). The sensitivity for the detection of inward and outward ALT-TFs was 0.80-0.96 and 0.66-0.93 (see Figure below). The precision was 1 across all simulated sequencing depths, indicating that TelFusDetector did not call false positive ALT-TFs in any of the normal samples.

Performance of TelFusDetector across different depths of coverage for inward and outward telomere fusions separately.

Next, we assessed the ability of TelFusDetector to identify the correct breakpoint sequence for each fusion. To this aim, we compared using the Levenshtein distance the raw and corrected breakpoint sequences by TelFusDetector with the true breakpoint sequences of the 184 simulated fusions using the T2T-CHM13 assembly. These results showed (see Figure below) that the correction of the breakpoint sequence (red dots) included in our algorithm achieved high similarity (median ~ 1) with the real breakpoint sequences, whereas the raw breakpoint sequences were more dissimilar to the true sequences due to sequencing errors.

Levenshtein distance between the expected and observed breakpoint sequences of the telomere fusions detected by TelFusDetector. Dots correspond to the breakpoint sequence from a telomere fusion.

2. The various TF-rate across cancers should be discussed in more details. For example, some cancer types are enriched with TF, like CNS tumors and sarcoma. Is this aligned with previous results in the field? What is the relationship between TF-rate and SV occurrence rate and mutations such as TP53?
3. More analyses on the association of the TF rate with other features relevant to chromosome instability, such as chromothripsis, breakage-fusion-bridge and the number of different patterns of SV (PMID: 32025012), should be analyzed. Some telomere-relevant features like TERT expression and TRF1/2 expression should also be included.

The rate of ALT-TFs correlated with the rate of ALT-positive tumours per cancer type (panel A in the Figure below). These results are comparable for both inward and outward ALT-TF (panels B and C in the Figure below).

Correlation of the median telomere fusion rates and the fraction of ALT+ samples per cancer type for all (A), inward (B) and outward (C) telomere fusion rates.

As can be seen in the Figure below extracted from Sieverling et al. 2020 (PMID: 32024817), osteosarcomas, pancreatic neuroendocrine tumours and CNS tumours show the highest frequencies of ALT-associated mutations (e.g., inactivating mutations in *ATRX/DAXX*), which is consistent with the high frequency of ALT-TFs observed in these tumour types in our analysis.

Figure 1b extracted from Sieverling et al 2020 (PMID: 32024817). Distribution of TMM-associated mutations across the tumour types analysed in PCAWG.

Next, we assessed the association of inactivating mutations in *TP53* and the SV rate with the ALT-TF rates. In addition to these variables, we also included the presence of chromosome instability features, ALT status and other biological and genomic features (see description in the next section) as covariates. Specifically, we expanded the number of variables included in the association analysis. Specifically, we included donor sex, ploidy, age at diagnosis, ALT status, WGD status, *TERT*-associated mutations, telomere content, presence of chromothripsis events, truncating mutations in *TP53*, number of SVs per sample, chromosome instability signatures (BFB_cycles, chromatin_bridge, ecDNA, DM, HSR, Hourglass, Large_gain, Large_loss and Micronuclei; PMID: 35835961), and viral infection information (Alphapapillomavirus, Alphatorquevirus, Cytomegalovirus, Gammaretrovirus, Lentivirus, Lymphocryptovirus, Orthohepadnavirus and Roseolovirus; PMID: 32025001).

We first used a linear model to regress out the effect of the cancer type on the rate of ALT-TFs. We limited this analysis to tumour samples from the PCAWG project with a tumour purity of at least 30%. The residuals of the linear model were then used in a second linear model that incorporated all the variables previously mentioned. We then performed backward stepwise model selection using the Akaike information criterion to identify covariates associated with ALT-TF formation. After applying backward stepwise model selection using the Akaike information criterion to identify covariates associated with ALT-TF formation, we found that in addition to ALT status and telomere content, truncating mutations in *TP53* also showed a significant association with the rate of inward and outward ALT-TFs (see Figure below). Genomic instability signatures, as well as the other biological and genomic features

described above, were not significantly associated with the rate of ALT-TFs in the PCAWG cohort. We have now included these results in the manuscript.

Coefficient values estimated using linear regression analysis and variable selection for the covariates with the strongest positive and negative association with inward (left) and outward (right) ALT-TFs.

We first used a linear model to regress out the effect of the cancer type on the rate of ALT-TFs. We limited this analysis to tumour samples from the PCAWG project with a tumour purity of at least 30%. The residuals of the linear model were then used in a second linear model that incorporated all the variables previously mentioned. We then performed backward stepwise model selection using the Akaike information criterion to identify covariates associated with ALT-TF formation. As shown in the plot below, only the ALT-status, TP53 mutations, and telomere content were positively correlated with the ALT-TFs. In particular, we observed a very strong correlation between outward TFs and ALT-status. In contrast, only activating mutations or SVs in TERT were negatively correlated with the TF formation. None of the chromosome instability features included in the model were significantly associated with the TF formation, suggesting biological processes behind TF and other types of SV events.

Coefficient values estimated using linear regression analysis and variable selection for the covariates with the strongest positive and negative association with TF rates for inward (left) and outward (right) telomere fusions.

Finally, and as suggested by the reviewer, we evaluated the importance of the expression of some relevant genes (*ATRX*, *TRF1/2*, *TERT*, *DAXX* and *TERRA*) in combination with other biological factors. After normalization, we observed that only *TERRA* expression levels significantly and positively correlated with the TF-rates in cancer (see plot below).

Coefficient values estimated using linear regression analysis and variable selection for the covariates (levels of expression per gene) with the strongest positive and negative association with TF rates for inward (left) and outward (right) telomere fusions.

4. When calculating the TF rate, the authors divided the number of fusion reads by tumor purity. Since the estimation of tumor purity, either by histological review or by computational estimation, is not very accurate, such normalization is too dependent on purity and may lead to unexpected bias. I would suggest the authors to:

a) Show the correlations of TF rates with tumor purity before and after the purity correction to demonstrate the necessity and effect of the normalization.

b) First categorize the samples to different groups based on their purity (e.g., high, middle, low) and then repeat analyses in Fig. 2 using the raw TF rate without purity correction.

Please see our response in General Comments.

5. Another question about purity. When the authors analyzed the genomic correlates of ALT-TF, they used “linear model to regress out the effect of the cancer type and tumour purity on the rate of ALT-TFs”. However, if I understand correctly, the purity has already been considered when calculating the “rate of ALT-TFs”. Why is it regressed again in the model?

Thanks to the reviewer for pointing out this issue. We have now repeated the analysis without including purity as a covariate in the linear model. As observed in the previous sections, the results remain unchanged. Please see our response in General Comments.

6. The authors successfully applied this method to liquid biopsy. However, as a biomarker for cancer detection, TFs should be compared with other biomarkers that can be inferred from ctDNA, such as fragmentome (PMID: 31142840, 34417454), presence of oncogenic mutations, etc.

Please see our response in General Comments. We appreciate the reviewer's suggestion. Unfortunately, the PCAWG database does not include ctDNA data, and although we have gathered ctDNA samples from human glioblastoma patients, the number of samples (n=8), is insufficient for meaningful comparative studies. However, we envision that by integrating ALT-TF information with fragmentomics, presence of oncogenic mutations, and other data obtained by the analysis of cell-free DNA from cancer patients, the predictive power for detecting cancer through liquid biopsy analysis will be enhanced. We have highlighted this point in the discussion section of the manuscript: "Enrichment and amplification..., thus complementing current biopsy methods".

7. I would recommend the authors to add more explanations about the TFs identified in normal tissues and blood cells. For example, in Fig. 4D (PD54225), there are significantly more TFs in whole blood than serum, indicating they were derived from normal blood cell. Is this biologically meaningful or only technical noise? Is there any way to remove them and retain only somatic TFs?

We thank the Reviewer for bringing up this point. For a given fusion and using short reads, we cannot fully ascertain whether it corresponds to a true somatic fusion occurring in e.g. a blood cell or technical noise. However, we can determine which properties of the fusions are enriched in tumours or in the blood of individuals with cancer as compared to normal blood or tissues using the Random Forest model that we trained to distinguish blood from healthy controls and individuals with cancer. This analysis revealed that the rate of pure ALT-TFs is enriched in the blood of individuals with cancer. These results are presented in Supplementary Figure 14C (also copied below).

Mean decrease in accuracy (importance) obtained for each variable used in the Random Forest model. The top 20 predictive features are shown.

Minor concerns:

1. Figure 1B. If I understand correctly, each bar in the plot corresponds to the TF rate of one patient and the bars are ordered decreasingly by the TF rates. This should be clarified in the paper.

We have now clarified this point in the manuscript.

2. Definitions of some terminologies should be clarified for easier understanding, such as telomere content, TERTp as in Figure 2 and C-circle.

We have now clarified this point in the manuscript.

3. Figure 2, colors in many boxplots and barplots are hardly visible. Please consider replot (maybe in log scale)?

We thank the Reviewer for this suggestion. We have tried multiple ways to visualise the data reported in Figure 2. We have found that the way in which we report the data is the best compromise we can find. We recognize that due to the small range of values shown in some cases the boxes of the box plots are very thin. However, as the colour of the boxplots is consistent across cancer types, we consider that the groups can be easily identified.

4. The genes used for evaluating ALT activity should be given in a supplementary table. Page 4, “when the TERRA-AS not treated with RNase H or the TERRA sense transcript (TERRA-S) were used” should be “when the TERRA-AS was not treated with RNase H or the TERRA sense transcript (TERRA-S) was used”?

We thank the Reviewer for this suggestion. We have edited the manuscript accordingly. We used the ALT classification status for all samples provided by Sieverling et al 2020 (PMID: 32024817) and de Nonneville & Reddel 2020 (PMID: 33692341).

REVIEWERS' COMMENTS

Reviewer #1 (Remarks to the Author):

The revised manuscript has addressed most questions, yet there remains a concern.

I suggest refining the text related to Zscan4 and ALT in mES cells on Page 5. It is important to note that mESCs lack conventional ALT-associated features such as APBs. Additionally, the reference cited in the main text is a review article that does not furnish any empirical support for the connection between ALT features and Zscan4 in mES cells. The assertion presented in this context appears to be overstated. The work reported by Zalzman et al. in Nature 2010 shows that Zscan4 regulates telomere elongation in mES cells, and that overexpressing Zscan4 promotes telomere sister chromatid exchange. However, whether this observed sister chromatid exchange at telomeres in mES cells is linked to Alternative lengthening of Telomeres remains elusive. I would recommend that the authors focus on the connection between the observation of telomere sister chromatid exchange in mES cells and telomere fusion events rather than ALT features.

Reviewer #2 (Remarks to the Author):

My points have been addressed. I have no further comments.

Reviewer #3 (Remarks to the Author):

My concerns have been addressed and I have no further comments.

Response to Reviewers

Point-by-Point Responses

Reviewer #1 (Remarks to the Author):

The revised manuscript has addressed most questions, yet there remains a concern.

I suggest refining the text related to Zscan4 and ALT in mES cells on Page 5. It is important to note that mESCs lack conventional ALT-associated features such as APBs. Additionally, the reference cited in the main text is a review article that does not furnish any empirical support for the connection between ALT features and Zscan4 in mES cells. The assertion presented in this context appears to be overstated. The work reported by Zalzman et al. in Nature 2010 shows that Zscan4 regulates telomere elongation in mES cells, and that overexpressing Zscan4 promotes telomere sister chromatid exchange. However, whether this observed sister chromatid exchange at telomeres in mES cells is linked to Alternative lengthening of Telomeres remains elusive. I would recommend that the authors focus on the connection between the observation of telomere sister chromatid exchange in mES cells and telomere fusion events rather than ALT features.

We thank the reviewer for the suggestion. In the revised manuscript, we have refined the text in order to reflect the connection between ALT-TF and the telomere sister chromatid exchange (T-SCE) mechanism described in mES cells. Additionally, we have changed the reference to Zalzman et al. Nature 2010 (PMID: 20336070)

The revised text now reads as follows:

“Furthermore, these findings indicate the presence of ALT-TF within mESCs, a cell type that not only exhibits telomere activity but possesses the capacity to elongate its telomeres through a telomere sister chromatid exchange (T-SCE) mechanism (Zalzman et al. Nature 2010)”

Reviewer #2 (Remarks to the Author):

My points have been addressed. I have no further comments.

Reviewer #3 (Remarks to the Author):

My concerns have been addressed and I have no further comments.